# A Stochastic Approximation Approach for Efficient Decentralized Optimization on Random Networks

## Abstract

1      A challenging problem in decentralized optimization is to develop algorithms with
2      fast convergence on random and time varying topologies under unreliable and
3      bandwidth-constrained communication network. This paper studies a stochastic
4      approximation approach with a Fully Stochastic Primal Dual Algorithm (FSPDA)
5      framework. Our framework relies on a novel observation that randomness in time
6      varying topology can be incorporated in a stochastic augmented Lagrangian for-
7      mulation, whose expected value admits saddle points that coincide with stationary
8      solutions of the decentralized optimization problem. With the `FSPDA` framework,
9      we develop two new algorithms supporting efficient sparsified communication on
10      random time varying topologies — `FSPDA-SA` allows agents to execute multiple
11      local gradient steps depending on the time varying topology to accelerate conver-
12      gence, and `FSPDA-STORM` further incorporates a variance reduction step to improve
13      sample complexity. For problems with smooth (possibly non-convex) objective
14      function, within $T$ iterations, we show that `FSPDA-SA` (resp. `FSPDA-STORM`) finds
15      an $\mathcal{O}(1/\sqrt{T})$-stationary (resp. $\mathcal{O}(1/T^{2/3})$) solution. Numerical experiments show
16      the benefits of the `FSPDA` algorithms.

## 1   Introduction

18 Consider $n$ agents that communicate on an undirected and connected graph/network $\mathcal{G} = (\mathcal{V}, \mathcal{E})$ with
19 $\mathcal{V} = [n] := \{1, \dots, n\}, \mathcal{E} \subseteq \mathcal{V} \times \mathcal{V}$. Each agent $i \in [n]$ has access to a continuously differentiable
20 (possibly non-convex) local objective function $f_i : \mathbb{R}^d \to \mathbb{R}$ and maintains a local decision variable
21 $\mathbf{x}_i \in \mathbb{R}^d$. Denote $\mathbf{x} = [\mathbf{x}_1^\top, ..., \mathbf{x}_n^\top]^\top \in \mathbb{R}^{nd}$. Our aim is to tackle:

$$\min_{\mathbf{x} \in \mathbb{R}^{nd}} \ \frac{1}{n} \sum_{i=1}^{n} f_i(\mathbf{x}_i) \quad \text{s.t.} \quad \mathbf{x}_i = \mathbf{x}_j, \ \forall \ (i, j) \in \mathcal{E}. \tag{1}$$

22 In other words, (1) seeks a $\mathbf{x}^\star \in \mathbb{R}^d$ that minimizes $F(\mathbf{x}) := (1/n) \sum_{i=1}^{n} f_i(\mathbf{x})$. We are interested
23 in the stochastic optimization setting where each $f_i(\mathbf{x}_i)$ is given by (with slight abuse of notation)

$$f_i(\mathbf{x}_i) := \mathbb{E}_{\xi_i \sim \mathbb{P}_i}[f_i(\mathbf{x}_i; \xi_i)] \tag{2}$$

24 where $\mathbb{P}_i$ represents the $i$-th data distribution. Problem (1) is relevant to the distributed learning
25 problem especially in the decentralized case where a central server is absent. Prior works [Nedic and
26 Ozdaglar, 2009, Lian et al., 2017, Nedic et al., 2017, Qu and Li, 2017] demonstrated that *decentralized*
27 algorithms can tackle (1) efficiently through repeated message exchanges among the neighbors and
28 local stochastic gradient updates.

29 Towards an efficient decentralized algorithm for (1), an important direction is to consider a *time*
30 *varying graph topology* setting where the *active edge set* in $\mathcal{G}$ changes over time. This is a generic
31 setting covering cases when the communication links are unreliable, or the agents choose not to
32 communicate in a certain round (a.k.a. local updates) [Koloskova et al., 2019a, Nadiradze et al., 2021].

| Prior Works | SG | TV | w/o BH | Rate |
|---|---|---|---|---|
| Prox-GPDA [Hong et al., 2017] | ✗ | ✗ | ✓ | Asympt. |
| NEXT [Lorenzo and Scutari, 2016] | ✗ | ✓ | ✓ | Asympt. |
| DSGD [Koloskova et al., 2020] | ✓ | ✓ | ✗ | $\mathcal{O}(\sigma/\sqrt{nT})$ |
| Swarm-SGD [Nadiradze et al., 2021] | ✓ | ✓ | ✗ | $\mathcal{O}(\sigma^2/\sqrt{T})$ |
| CHOCO-SGD [Koloskova et al., 2019a] | ✓ | ✗$^{\ddagger}$ | ✗ | $\mathcal{O}(\sigma/\sqrt{nT})$ |
| Decen-Scaffnew [Mishchenko et al., 2022] | ✓ | ✗$^{\dagger}$ | ✓ | $\mathcal{O}(\sigma/\sqrt{nT})$ |
| Local-GT [Liu et al., 2024] | ✓ | ✗$^{\dagger}$ | ✓ | $\mathcal{O}(\sigma/\sqrt{nT})$ |
| LED [Alghunaim, 2024] | ✓ | ✗$^{\dagger}$ | ✓ | $\mathcal{O}(\sigma/\sqrt{nT})$ |
| FSPDA-SA (**This Work**) | ✓ | ✓ | ✓ | $\mathcal{O}(\sigma/\sqrt{nT})$ |
| FSPDA-STORM (**This Work**) | ✓ | ✓ | ✓ | $\mathcal{O}(\sigma^{2/3}/T^{2/3})$ |

Table 1: Comparison of decentralized algorithms for **non-convex** optimization. In the table, 'SG' is 'Stochastic Gradient', 'TV' is 'Time Varying Graph', 'w/o BH' is 'Without Bounded Heterogeneity', and 'Rate' is the expected squared gradient norm $\mathbb{E}[\|\nabla F(\bar{\mathbf{x}})\|^2]$ after $T$ iterations. Note that $\sigma^2$ is the variance of stochastic gradient. $^{\ddagger}$CHOCO-SGD incorporates broadcast gossip as a special case of compression. $^{\dagger}$ProxSkip, Local-GT, LED consider local updates with periodic communication.

By assuming that a random topology is drawn at each iteration, the convergence of decentralized stochastic gradient (DSGD) has been studied in [Lobel and Ozdaglar, 2010, Nadiradze et al., 2021] and is later on unified by [Koloskova et al., 2020] with tighter bounds for local updates, periodic sampling, etc. An alternative [Ram et al., 2010] is to analyze DSGD for the $B$-connectivity setting which requires the union of every $B$ consecutive time varying topologies to yield a connected graph. Nevertheless, these works focused on vanilla DSGD that may have slow convergence (in transient stage) and is limited to bounded data heterogeneity. The prior restrictions can be relaxed using advanced algorithms such as gradient tracking [Qu and Li, 2017], EXTRA [Shi et al., 2015] and primal-dual framework [Hong et al., 2017, Hajinezhad and Hong, 2019, Yi et al., 2021].

As noted by [Koloskova et al., 2021], analyzing the convergence of sophisticated algorithms with time varying topology, such as gradient tracking [Qu and Li, 2017] is challenging due to the non-symmetric product of two (or more) mixing matrices. Existing works considered various restrictions on the time varying topology $\mathcal{G}^{(t)} = (\mathcal{V}, \mathcal{E}^{(t)})$ and/or the problem (1): [Koloskova et al., 2021, Liu et al., 2024] studied gradient tracking with local updates that essentially takes $\mathcal{E}^{(t)} = \mathcal{E}$ periodically and $\mathcal{E}^{(t)} = \emptyset$ otherwise, also see [Mishchenko et al., 2022, Guo et al., 2023, Alghunaim, 2024] for a similar result and note that such algorithms require extra synchronization overhead; [Kovalev et al., 2021, 2024] considered a setting where $\mathcal{G}^{(t)}$ is connected for any $t$; [Nedic et al., 2017, Li and Lin, 2024] focused on (accelerated) gradient tracking with deterministic gradient when $F(\mathbf{x})$ is (strongly) convex; [Lorenzo and Scutari, 2016] also considered deterministic gradient with possibly non-convex $F(\mathbf{x})$ but only provides asymptotic convergence guarantees; [Lei et al., 2018, Yau and Wai, 2023] considered asymptotic convergence guarantees in the case of strictly (or strongly) convex $F(\mathbf{x})$. We provide a non-exhaustive list summarizing the convergence of existing works in Table 1.

The above discussion highlights a gap in the existing literature —

*Is there any algorithm that achieves fast convergence on time varying (random) topology?*

This paper gives an affirmative answer through developing the Fully Stochastic Primal Dual Algorithm (FSPDA) framework that leads to efficient decentralized algorithms tackling (1) in its general form. The framework features the design of a new stochastic augmented Lagrangian function.

As pointed out by [Chang et al., 2020], many decentralized algorithms (including gradient tracking) can be interpreted as primal-dual algorithms finding a saddle point of the augmented Lagrangian function. However, its extension to time varying topology is not straightforward due to the inconsistency in dual variables updates. To overcome this challenge, we propose a stochastic equality constrained reformulation of (1) to model randomness in topology. Then, the latter yields a stochastic augmented Lagrangian function. Applying stochastic approximation (SA) to solve the latter leads to the FSPDA framework. Our contributions are

- We propose two new algorithms: (i) FSPDA-SA is derived by vanilla SA that applies primal-dual stochastic gradient descent-ascent on the stochastic augmented Lagrangian, (ii) FSPDA-STORM uses an additional control variate / momentum term to reduce the drift term's variance in a recursive manner. Both algorithms are fully stochastic as the random time varying topology is treated as a part of randomness. Additionally, our framework supports sparsified communication, i.e., the agents can choose to communicate a subset of primal coordinates at each iteration.

- We show that after $T$ iterations, FSPDA-SA (resp. FSPDA-STORM) finds in expectation a solution whose squared gradient norm is $\mathcal{O}(1/\sqrt{T})$ (resp. $\mathcal{O}(1/T^{2/3})$). The convergence analysis is derived from a new Lyapunov function design that involves an unsigned inner product term and incorporates a variance condition on the random time varying topologies. Interestingly, we show empirically that using momentum in dual updates benefits the consensus error convergence.

- We also demonstrate that both FSPDA-SA and FSPDA-STORM can be implemented in a fully asynchronous manner, i.e., the agents can communicate and compute at different time slots, and supports local update as the algorithms allow for arbitrary time varying topology. That said, we remark that the convergence rates with local updates of FSPDA-SA and FSPDA-STORM are only suboptimal.

We provide numerical experiments to show that FSPDA-SA and FSPDA-STORM outperform existing algorithms in terms of iteration and communication complexity.

**Notations.** Let $\mathbf{W} \in \mathbb{R}^{d \times d}$ be a symmetric (not necessarily positive semidefinite) matrix, the $\mathbf{W}$-weighted (semi) inner product of vectors $\mathbf{a}, \mathbf{b} \in \mathbb{R}^d$ is denoted as $\langle \mathbf{a} \mid \mathbf{b} \rangle_{\mathbf{W}} := \mathbf{a}^\top \mathbf{W} \mathbf{b}$. Similarly, the $\mathbf{W}$-weighted (semi) norm is denoted by $\|\mathbf{a}\|_{\mathbf{W}}^2 := \langle \mathbf{a} \mid \mathbf{a} \rangle_{\mathbf{W}}$. The subscript notation is omitted for $\mathbf{I}$-weighted inner products. For any square matrix $\mathbf{X}$, $(\mathbf{X})^\dagger$ denotes its pseudo inverse.

# 2 The Fully Stochastic Primal Dual Algorithm (FSPDA) Framework

This section develops the FSPDA framework for tackling (1) and describes two variants of the framework leading to decentralized stochastic optimization of (1). Let $\widetilde{\mathbf{A}} \in \{-1, 0, 1\}^{|\mathcal{E}| \times n}$ be an incidence matrix of $\mathcal{G}$. By defining $\mathbf{A} = \widetilde{\mathbf{A}} \otimes \mathbf{I}_d \in \{-1, 0, 1\}^{|\mathcal{E}|d \times nd}$, we observe that the consensus constraint in (1) is equivalent to $\mathbf{A}\mathbf{x} = \mathbf{0}$.

Our first step is to model the randomness in the time varying topology using the random variable (r.v.) $\xi_a \sim \mathbb{P}_a$. For each realization $\xi_a$, we define the random incidence matrix $\mathbf{A}(\xi_a) := \mathbf{I}(\xi_a)\mathbf{A} \in \{-1, 0, 1\}^{|\mathcal{E}|d \times nd}$ where $\mathbf{I}(\xi_a) \in \{0, 1\}^{|\mathcal{E}|d \times |\mathcal{E}|d}$ is a binary diagonal matrix. In addition to selecting each edge of $\mathcal{G}$ randomly, $\mathbf{I}(\xi_a)$ selects a random subset of $d$ coordinates. As we will see later, this allows our approach to simultaneously achieve random sparsification for communication compression.

Assume that $\mathbb{E}_{\xi_a \sim \mathbb{P}_a}[\mathbf{I}(\xi_a)]$ is a positive diagonal matrix, (1) is equivalent to:

$$\min_{\mathbf{x} \in \mathbb{R}^{nd}} \ \tfrac{1}{n} \sum_{i=1}^n \mathbb{E}_{\xi_i \sim \mathbb{P}_i}[f_i(\mathbf{x}_i; \xi_i)] \quad \text{s.t.} \quad \mathbb{E}_{\xi_a \sim \mathbb{P}_a}[\mathbf{A}(\xi_a)]\mathbf{x} = \mathbf{0}. \tag{3}$$

Denote $\xi = (\xi_1, \ldots, \xi_n, \xi_a)$, FSPDA hinges on the following *augmented Lagrangian* function of (3):

$$\mathcal{L}(\mathbf{x}, \boldsymbol{\lambda}) := \mathbb{E}_\xi[\mathcal{L}(\mathbf{x}, \boldsymbol{\lambda}; \xi)]$$
$$\text{with } \mathcal{L}(\mathbf{x}, \boldsymbol{\lambda}; \xi) := \sum_{i=1}^n f_i(\mathbf{x}_i; \xi_i) + \tilde{\eta} \langle \boldsymbol{\lambda} \mid \mathbf{A}(\xi_a)\mathbf{x} \rangle + \tfrac{\tilde{\gamma}}{2} \|\mathbf{A}(\xi_a)\mathbf{x}\|^2, \tag{4}$$

where $\tilde{\eta} > 0, \tilde{\gamma} > 0$ are penalty parameters. It can be verified that the saddle points of $\mathcal{L}(\mathbf{x}, \boldsymbol{\lambda})$ correspond to the KKT points of (1) [Bertsekas, 2016]. For brevity, in the rest of this paper, we may drop the subscript in $\xi$ whenever the notation is clear from the context.

FSPDA is developed from applying stochastic approximation (SA) to seek a saddle point of (4). By recognizing $\mathbf{A}(\xi)^\top \mathbf{A}(\xi) = \mathbf{A}^\top \mathbf{A}(\xi)$, we consider the stochastic gradients:

$$\nabla_{\mathbf{x}}\mathcal{L}(\mathbf{x}, \boldsymbol{\lambda}; \xi) := \nabla \mathbf{f}(\mathbf{x}; \xi) + \tilde{\eta}\mathbf{A}^\top \boldsymbol{\lambda} + \tilde{\gamma}\mathbf{A}^\top \mathbf{A}(\xi)\mathbf{x}, \ \ \nabla_{\boldsymbol{\lambda}}\mathcal{L}(\mathbf{x}, \boldsymbol{\lambda}; \xi) := \tilde{\eta}\mathbf{A}(\xi)\mathbf{x}, \tag{5}$$

where $\nabla \mathbf{f}(\mathbf{x}; \xi) = [\nabla f_1(\mathbf{x}_1; \xi_1); \ldots; \nabla f_n(\mathbf{x}_n; \xi_n)] \in \mathbb{R}^{nd}$. Notice that to facilitate algorithm development, we have taken a deterministic $\mathbf{A}$ for the term in $\nabla_{\mathbf{x}}\mathcal{L}$ related to $\boldsymbol{\lambda}$. Now observe the $i$th $d$-dimensional block of $\mathbf{A}^\top \mathbf{A}(\xi)\mathbf{x}$ which can be aggregated within $\mathcal{N}_i(\xi)$ the neighborhood of the $i$th agent as:

$$\left[\mathbf{A}^\top \mathbf{A}(\xi)\mathbf{x}\right]_i = \sum_{j \in \mathcal{N}_i(\xi)} \mathbf{C}_{ij}(\xi)(\mathbf{x}_j - \mathbf{x}_i), \tag{6}$$

where $\mathbf{C}_{ij}(\xi) \in \{0, 1\}^{d \times d}$ is diagonal and depends on the selected coordinates for the edge $(i, j)$ under randomness $\xi$. Eq. (6) *only* relies on $\mathbf{x}_j$ from neighbor $j$ that is connected on the time varying

topology $\mathcal{G}(\xi)$. For illustration, an example of the above random graph model is given by Figure 3 in Appendix A. Importantly, (5) shows that with the stochastic augmented Lagrangian function, the time varying topology can be treated implicitly as a part of the randomness in the stochastic primal-dual gradients. The framework is thus described as being *fully stochastic* as in [Bianchi et al., 2021], and departs from [Liu et al., 2024, Alghunaim, 2024] that treat the topology as fixed during the derivation of primal-dual algorithm(s). From (5), (6), we derive *two* variants of FSPDA.

FSPDA-SA **Algorithm.** The first variant of FSPDA is derived from a direct application of stochastic gradient descent-ascent (SGDA) updates. Take $\alpha > 0, \beta > 0$ as the step sizes, we have

$$\mathbf{x}^{t+1} = \mathbf{x}^t - \alpha\nabla_{\mathbf{x}}\mathcal{L}(\mathbf{x}^t, \boldsymbol{\lambda}^t; \xi^t), \ \ \boldsymbol{\lambda}^{t+1} = \boldsymbol{\lambda}^t + \beta\nabla_{\boldsymbol{\lambda}}\mathcal{L}(\mathbf{x}^t, \boldsymbol{\lambda}^t; \xi^t). \tag{7}$$

Taking the variable substitution $\widehat{\boldsymbol{\lambda}} := \mathbf{A}^\top\boldsymbol{\lambda}$ yields the following recursion:

---

FSPDA-SA: for any $t \geq 0$ and any $i \in [n]$,

$$\mathbf{x}_i^{t+1} = \mathbf{x}_i^t - \alpha\nabla f_i(\mathbf{x}_i^t; \xi_i^t) - \eta\widehat{\boldsymbol{\lambda}}_i^t + \gamma\sum_{j\in\mathcal{N}_i(\xi_a^t)}\mathbf{C}_{ij}(\xi_a^t)(\mathbf{x}_j^t - \mathbf{x}_i^t), \tag{8a}$$

$$\widehat{\boldsymbol{\lambda}}_i^{t+1} = \widehat{\boldsymbol{\lambda}}_i^t + \beta\sum_{j\in\mathcal{N}_i(\xi_a^t)}\mathbf{C}_{ij}(\xi_a^t)(\mathbf{x}_j^t - \mathbf{x}_i^t). \tag{8b}$$

---

Note that $\mathbf{x}^0, \widehat{\boldsymbol{\lambda}}^0$ can be initialized arbitrarily.

FSPDA-STORM **Algorithm.** The second variant of FSPDA reduces the variance of the stochastic gradient term in (5) using the recursive momentum variance reduction technique [Cutkosky and Orabona, 2019]. Herein, the key idea is to utilize a control variate in estimating the (primal-dual) gradients of $\mathcal{L}(\mathbf{x}, \boldsymbol{\lambda})$. Take $\alpha, \beta > 0$ and $a_x, a_\lambda \in [0, 1]$ as the momentum parameters, we have $\mathbf{x}^{t+1} = \mathbf{x}^t - \alpha\mathbf{m}_x^t, \boldsymbol{\lambda}^{t+1} = \boldsymbol{\lambda}^t + \beta\mathbf{m}_\lambda^t$ as the primal-dual updates, and

$$\begin{aligned} \mathbf{m}_x^{t+1} &= \nabla_{\mathbf{x}}\mathcal{L}(\mathbf{x}^{t+1}, \boldsymbol{\lambda}^{t+1}; \xi^{t+1}) + (1 - a_x)(\mathbf{m}_x^t - \nabla_{\mathbf{x}}\mathcal{L}(\mathbf{x}^t, \boldsymbol{\lambda}^t; \xi^{t+1})), \\ \mathbf{m}_\lambda^{t+1} &= \nabla_{\boldsymbol{\lambda}}\mathcal{L}(\mathbf{x}^{t+1}, \boldsymbol{\lambda}^{t+1}; \xi^{t+1}) + (1 - a_\lambda)(\mathbf{m}_\lambda^t - \nabla_{\boldsymbol{\lambda}}\mathcal{L}(\mathbf{x}^t, \boldsymbol{\lambda}^t; \xi^{t+1})). \end{aligned} \tag{9}$$

The aim of $\mathbf{m}_x^{t+1}$ is to estimate $\nabla_{\mathbf{x}}\mathcal{L}(\mathbf{x}^{t+1}, \boldsymbol{\lambda}^{t+1})$. Now, instead of the straightforward estimator $\nabla_{\mathbf{x}}\mathcal{L}(\mathbf{x}^{t+1}, \boldsymbol{\lambda}^{t+1}; \xi^{t+1})$, we include an extra zero-mean term $\mathbf{m}_x^t - \nabla_{\mathbf{x}}\mathcal{L}(\mathbf{x}^t, \boldsymbol{\lambda}^t; \xi^{t+1})$ to reduce the variance of the stochastic gradient estimation. The latter is a control variate that is computed recursively. Particularly, it has been shown in [Cutkosky and Orabona, 2019] that it can effectively reduce variance with a carefully designed parameter $a_x$, provided that the stochastic gradient map satisfies a mean-square Lipschitz condition. We summarize the algorithm as follows.

---

FSPDA-STORM: for any $t \geq 0$ and any $i \in [n]$,

$$\mathbf{x}_i^{t+1} = \mathbf{x}_i^t - \alpha\mathbf{m}_{x,i}^t, \tag{10a}$$

$$\widehat{\boldsymbol{\lambda}}_i^{t+1} = \widehat{\boldsymbol{\lambda}}_i^t + \beta\mathbf{m}_{\lambda,i}^t, \tag{10b}$$

$$\mathbf{m}_{x,i}^{t+1} = (1 - a_x)\big[\mathbf{m}_{x,i}^t + \nabla f_i(\mathbf{x}_i^t; \xi_i^{t+1}) - \eta\widehat{\boldsymbol{\lambda}}_i^t + \gamma\sum_{j\in\mathcal{N}_i(\xi_a^{t+1})}\mathbf{C}_{ij}(\xi_a^{t+1})(\mathbf{x}_j^t - \mathbf{x}_i^t)\big] \tag{10c}$$

$$\quad + \nabla f_i(\mathbf{x}_i^{t+1}; \xi_i^{t+1}) - \eta\widehat{\boldsymbol{\lambda}}_i^{t+1} + \gamma\sum_{j\in\mathcal{N}_i(\xi_a^{t+1})}\mathbf{C}_{ij}(\xi_a^{t+1})(\mathbf{x}_j^{t+1} - \mathbf{x}_i^{t+1})$$

$$\mathbf{m}_{\lambda,i}^{t+1} = (1 - a_\lambda)\big[\mathbf{m}_{\lambda,i}^t + \sum_{j\in\mathcal{N}_i(\xi_a^{t+1})}\mathbf{C}_{ij}(\xi_a^{t+1})(\mathbf{x}_j^t - \mathbf{x}_i^t)\big] \tag{10d}$$

$$\quad + \sum_{j\in\mathcal{N}_i(\xi_a^{t+1})}\mathbf{C}_{ij}(\xi_a^{t+1})(\mathbf{x}_j^{t+1} - \mathbf{x}_i^{t+1})$$

---

Note that to achieve the theoretical performance (see later in Sec. 3), $\mathbf{x}^0, \widehat{\boldsymbol{\lambda}}^0, \mathbf{m}_x^0, \mathbf{m}_\lambda^0$ shall be initialized as $\mathbf{x}_i^0 = \bar{\mathbf{x}}^0$, $\widehat{\boldsymbol{\lambda}}_i^0 = (\alpha/\eta)n^{-1}(\nabla F(\bar{\mathbf{x}}^0) - \nabla f_i(\bar{\mathbf{x}}^0))$, $\mathbf{m}_{x,i}^0 = \nabla F(\bar{\mathbf{x}}^0)$, $\mathbf{m}_{\lambda,i}^0 = \mathbf{0}$ according to (23). We remark that a simple initialization choice $\widehat{\boldsymbol{\lambda}}^0 = \mathbf{m}_{x,i}^0 = \mathbf{m}_{\lambda,i}^0 = \mathbf{0}$ works well in practice.

Both FSPDA-SA and FSPDA-STORM are decentralized algorithms that can be implemented on random time varying topology, and support randomized sparisification for further communication compression. The key is to observe that in (8), (10), the only information required for agent $i$ is to obtain $\sum_{j\in\mathcal{N}_i(\xi_a^t)}\mathbf{C}_{ij}(\xi_a^t)(\mathbf{x}_j^t - \mathbf{x}_i^t)$, and in addition $\sum_{j\in\mathcal{N}_i(\xi_a^t)}\mathbf{C}_{ij}(\xi_a^t)(\mathbf{x}_j^{t-1} - \mathbf{x}_i^{t-1})$ for FSPDA-STORM, at iteration $t$.

## 2.1 Implementation Details and Connection to Existing Works

We discuss several features of the FSPDA algorithms and their connections to existing works.

**Local & Asynchronous Updates.** The *local update* scheme where each agent $i$ is allowed to update its own local variables $\mathbf{x}_i, \boldsymbol{\lambda}_i$ for multiple iterations without a communication step is a common practice in decentralized optimization [Liu et al., 2024, Li and Lin, 2024, Alghunaim, 2024, Mishchenko et al., 2022]. As discussed before, such scheme can be seen as a special case of the FSPDA framework where the time varying topology $\mathcal{E}^{(t)}$ is chosen such that the latter alternates between $\mathcal{E}^{(t)} = \mathcal{E}$ and $\mathcal{E}^{(t)} = \emptyset$.

Furthermore, FSPDA-SA allows for the general case of *asynchronous* updates. This is done so by taking the stochastic gradient as $\nabla f_i(\mathbf{x}_i^t; \xi^t) = b_i(\xi^t) \bar{b}_i \nabla f_i(\mathbf{x}_i^t; \xi^t)$ such that $b_i(\xi^t) \in \{0, 1\}$ with $\mathbb{E}[b_i(\xi^t)] = 1/\bar{b}_i$ for some constant $\bar{b}_i > 0$. Detailed discussions for a fully asynchronous implementation of FSPDA-SA can be found in Appendix A.

**Connection to Existing Works.** Evaluating $\mathbf{x}^{t+2} - \mathbf{x}^{t+1}$ from the FSPDA-SA sequence and observe that the combination of (8a) and (8b) is equivalent to the second order recursion:

$$
\begin{aligned}
\mathbf{x}^{t+2} = 2\left(\mathbf{I} - \frac{\gamma}{2}\mathbf{A}^\top\mathbf{A}(\xi^{t+1})\right)\mathbf{x}^{t+1} - \left(\mathbf{I} - (\gamma - \eta\beta)\mathbf{A}^\top\mathbf{A}(\xi^t)\right)\mathbf{x}^t \\
- \alpha\left(\nabla\mathbf{f}(\mathbf{x}^{t+1}; \xi^{t+1}) - \nabla\mathbf{f}(\mathbf{x}^t; \xi^t)\right).
\end{aligned}
\tag{11}
$$

This reduces the FSPDA-SA recursion into a primal-only sequence by eliminating the dual sequence $\boldsymbol{\lambda}^t$. In the deterministic optimization setting when $\mathbf{A}(\xi) \equiv \mathbf{A}$ and $\nabla\mathbf{f}(\mathbf{x}; \xi) \equiv \nabla\mathbf{f}(\mathbf{x})$, (11) is equivalent to the EXTRA algorithm [Shi et al., 2015] using the mixing matrix $\mathbf{W} = \mathbf{I} - \gamma\mathrm{Diag}(\tilde{\mathbf{W}}\mathbf{1}) + \gamma\tilde{\mathbf{W}}$ where $\tilde{\mathbf{W}}$ is the 0-1 adjacency matrix of $\mathcal{G}$. Here, with an appropriate choice of $\gamma$, $\mathbf{W}$ will be doubly stochastic and satisfies the convergence requirement in [Shi et al., 2015]. Similar observations have been made in [Nedic et al., 2017] for the gradient tracking and DIGing algorithms.

On the other hand, for stochastic optimization on random networks, (11) suggests each agent to keep the current and previous iterates received from neighbors in the corresponding time varying topology. In this case, (11) yields an extension of the EXTRA/GT algorithms to time varying topology.

## 3 Convergence Analysis of FSPDA

This section presents the convergence rate analysis of FSPDA for (1). Unless otherwise specified, we focus on the case with smooth but possibly non-convex objective function. Specifically, we consider:

**Assumption 3.1.** *Each $f_i$ is $L$-smooth, i.e., for $i = 1, \ldots, n$,*

$$
\|\nabla f_i(\mathbf{x}) - \nabla f_i(\mathbf{y})\| \leq L\|\mathbf{x} - \mathbf{y}\| \ \forall \ \mathbf{x}, \mathbf{y} \in \mathbb{R}^d.
\tag{12}
$$

*There exists $f_\star > -\infty$ such that $f_i(\mathbf{x}) \geq f_\star$ for any $\mathbf{x} \in \mathbb{R}^d$.*

Note this implies that the global objective function $F(\cdot)$ is $L$-smooth but possibly non-convex.

We further assume that the random network $\mathcal{G}(\xi_a)$ is connected in expectation, yet each realization $\mathcal{G}(\xi_a)$ may not be connected. Let $\mathbf{R} = \mathbb{E}[\mathbf{I}(\xi_a)]$, this leads to the following property concerning the expected graph Laplacian matrix $\mathbf{A}^\top\mathbf{R}\mathbf{A} = \mathbb{E}[\mathbf{A}(\xi_a)^\top\mathbf{A}]$. Defining the matrix $\mathbf{K} := (\mathbf{I}_n - \mathbf{1}\mathbf{1}^\top/n) \otimes \mathbf{I}_d$, we have

**Assumption 3.2.** *There exists $\rho_{\max} \geq \rho_{\min} > 0$ and $\bar{\rho}_{\max} \geq \bar{\rho}_{\min} > 0$ such that*

$$
\rho_{\min}\mathbf{K} \preceq \mathbf{A}^\top\mathbf{R}\mathbf{A} \preceq \rho_{\max}\mathbf{K} \quad and \quad \bar{\rho}_{\min}\mathbf{K} \preceq \mathbf{A}^\top\mathbf{A} \preceq \bar{\rho}_{\max}\mathbf{K}.
\tag{13}
$$

It holds that $\mathbf{A}^\top\mathbf{R}\mathbf{A}\mathbf{K} = \mathbf{A}^\top\mathbf{R}\mathbf{A} = \mathbf{K}\mathbf{A}^\top\mathbf{R}\mathbf{A}$. The above assumption can be satisfied if $\mathcal{G}$ is connected [Yi et al., 2021], [Yi et al., 2018, Lemma 2] and $\mathrm{diag}(\mathbf{R}) > \mathbf{0}$ such that each edge is selected with a positive probability. As an important consequence, if $\gamma \leq \rho_{\min}/\rho_{\max}^2$, we have

$$
\|(\mathbf{I} - \gamma\mathbf{A}^\top\mathbf{R}\mathbf{A})\mathbf{x}\|_\mathbf{K}^2 \leq (1 - \gamma\rho_{\min})\|\mathbf{x}\|_\mathbf{K}^2, \ \forall \ \mathbf{x} \in \mathbb{R}^{nd}.
$$

We thus observe that the operator $(\mathbf{I} - \gamma\mathbf{A}^\top\mathbf{R}\mathbf{A})$ serves a similar purpose as the mixing matrix in a average consensus algorithms and $\rho_{\min}$ can be interpreted as the spectral radius of $\mathcal{G}$ similar

to [Koloskova et al., 2020, Eq. (12)]. Moreover, if we define $\mathbf{Q} := (\mathbf{A}^\top \mathbf{R} \mathbf{A})^\dagger$ such that it holds $\mathbf{Q}\mathbf{A}^\top \mathbf{R}\mathbf{A} = \mathbf{A}^\top \mathbf{R}\mathbf{A}\mathbf{Q} = \mathbf{K}$, Assumption 3.2 implies that $\rho_{\max}^{-1}\mathbf{K} \preceq \mathbf{Q} \preceq \rho_{\min}^{-1}\mathbf{K}$.

Next we consider several assumptions on the noise variance of the random quantities in FSPDA:

**Assumption 3.3.** *For any fixed* $\mathbf{x}_i \in \mathbb{R}^d$, $i \in [n]$, *there exists* $\sigma_i \geq 0$ *such that*

$$\mathbb{E}_{\xi_i \sim \mathbb{P}_i}[\|\nabla f_i(\mathbf{x}_i; \xi_i) - \nabla f_i(\mathbf{x}_i)\|^2] \leq \sigma_i^2. \tag{14}$$

*To simplify notations, we define* $\bar{\sigma}^2 := (1/n)\sum_{i=1}^n \sigma_i^2$.

**Assumption 3.4.** *For any fixed* $\mathbf{x} \in \mathbb{R}^{nd}$, *there exists* $\sigma_A \geq 0$ *such that*

$$\mathbb{E}_{\xi_a \sim \mathbb{P}_a}[\|\mathbf{A}(\xi_a)^\top \mathbf{A}\mathbf{x} - \mathbf{A}^\top \mathbf{R}\mathbf{A}\mathbf{x}\|^2] \leq \sigma_A^2 \|\mathbf{x}\|_{\mathbf{K}}^2. \tag{15}$$

Assumption 3.3 is standard. Meanwhile for Assumption 3.4, the variance term $\sigma_A^2$ measures the quality of the random topology $\mathcal{G}(\xi_a)$ in approximating the expected graph Laplacian $\mathbf{A}^\top \mathbf{R}\mathbf{A}$. The latter is important as it contributes to the variance in the drift term of FSPDA. Observe that $\sigma_A^2$ decreases with the proportion of edges selected in each random subgraph $\mathcal{G}(\xi_a)$.

To facilitate our discussions, we define the following quanitites:

$$\bar{\mathbf{x}}^t := \tfrac{1}{n}\sum_{i=1}^n \mathbf{x}_i^t, \quad \sum_{i=1}^n \|\mathbf{x}_i^t - \bar{\mathbf{x}}^t\|^2 = \|\mathbf{x}^t\|_{\mathbf{K}}^2. \tag{16}$$

**Convergence of** FSPDA-SA**.** We summarize the convergence rate for FSPDA-SA as follows. The proof can be found in Appendix C:

> **Theorem 3.5.** *Under Assumptions 3.1, 3.2, 3.3, 3.4. Suppose that the step sizes satisfy the conditions defined in* (46)*. Then, for any* $T \geq 1$ *with the random stopping iteration* $\mathsf{T} \sim$ $\mathrm{Unif}\{0, ..., T-1\}$, *the iterates generated by* FSPDA-SA *satisfy*
>
> $$\mathbb{E}\left[\|\nabla F(\bar{\mathbf{x}}^{\mathsf{T}})\|^2\right] \leq \frac{F_0 - f_\star}{\alpha T/8} + 8\alpha \mathbb{C}_\sigma \frac{\bar{\sigma}^2}{n}, \tag{17}$$
>
> $$\mathbb{E}\left[\sum_{i=1}^n \|\mathbf{x}_i^{\mathsf{T}} - \bar{\mathbf{x}}^{\mathsf{T}}\|^2\right] \leq \frac{F_0 - f_\star}{\mathsf{a}\gamma\rho_{\min}T/8} + \frac{8\alpha^2 \mathbb{C}_\sigma \bar{\sigma}^2}{\mathsf{a}\gamma\rho_{\min}n}, \tag{18}$$
>
> *for any* $\mathsf{a} > 0$, *where* $F_0$, $\mathbb{C}_\sigma$ *are defined in* (44)*,* (50)*.*

Setting $\mathsf{a} = \mathcal{O}(n/\sqrt{T\bar{\sigma}^2})$, $\alpha = \sqrt{n/(T\bar{\sigma}^2)}$ (and assuming $\bar{\sigma} > 0$), we have

$$\mathbb{E}\left[\|\nabla F(\bar{\mathbf{x}}^{\mathsf{T}})\|^2\right] = \mathcal{O}\left(\bar{\sigma}/\sqrt{nT}\right), \tag{19}$$

which is the same *asymptotic convergence rate* as a centralized SGD algorithm that takes $n$ stochastic gradient samples uniformly from each agent, i.e., linear speedup [Lian et al., 2017]. Also, using $\mathsf{a} = 1$, the consensus error converges as a rate of $\mathbb{E}\left[\sum_{i=1}^n \|\mathbf{x}_i^{\mathsf{T}} - \bar{\mathbf{x}}^{\mathsf{T}}\|^2\right] = \mathcal{O}(n^2\sigma_A^2\rho_{\max}/(T\rho_{\min}^2))$ under the same step size choice used in (19). Notice that for $T \gg 1$, the effect of random topology only degrades the convergence of consensus error, keeping the transient rate in (19) unaffected. If the gradients are deterministic ($\bar{\sigma} = 0$), setting $\mathsf{a} = (L^2\eta_\infty\rho_{\min})^{1/3}$, $\alpha = \alpha_\infty$ will yield a better convergence rate as $\mathbb{E}\left[\|\nabla F(\bar{\mathbf{x}}^{\mathsf{T}})\|^2\right] = \mathcal{O}(\sigma_A^4\sqrt{n}/T)$. Without a transient phase, the error due to random graph and coordinate sparsification is persistent through $\sigma_A^4$ in the above convergence rate.

We further show that the convergence of FSPDA-SA can be accelerated if the objective function of (1) satisfies the Polyak-Lojasiewicz (PL) condition:

**Assumption 3.6.** *There exists a constant* $\mu > 0$ *such that* $2\mu(F(\mathbf{x}) - f_\star) \leq \|\nabla F(\mathbf{x})\|^2$, $\forall \mathbf{x} \in \mathbb{R}^d$.

Assumption 3.6 includes strongly convex functions as a special case, but also includes other non-convex functions; see [Karimi et al., 2016]. We observe:

> **Corollary 3.7.** *Suppose the assumptions and step size conditions in Theorem 3.5 hold. Furthermore, with Assumption 3.6, there exists* $\delta \in (0, 1)$ *such that for any* $t \geq 0$,
>
> $$\mathbb{E}_t[F_{t+1} - f_\star] \leq (1 - \delta)(F_t - f_\star) + \mathbb{C}_\sigma\alpha^2\bar{\sigma}^2/n \tag{20}$$
>
> *for* $F_t$, $\mathbb{C}_\sigma$ *defined in* (44)*,* (70)*, and* $\delta = \min\{\alpha\mu/4,\ \gamma\rho_{\min}/16,\ \eta\beta/(3\rho_{\min}), \eta/12\}$.

The proof can be found in Appendix C.6. By setting $\alpha = c\ln(T)/(n^2 T)$ in (20), with a carefully chosen $c$ and a sufficiently large $T$ such that $\alpha \leq \alpha_\infty$, we can ensure that

$$\mathbb{E}\left[F(\bar{\mathbf{x}}^T) - f_\star + \|\mathbf{x}^T\|_{\mathbf{K}}^2\right] = \mathcal{O}\left(\bar{\sigma}^2 \ln(T)/(\mu n T)\right) \tag{21}$$

In the case of deterministic gradient, i.e., $\bar{\sigma}^2 = 0$, by setting $\alpha = \alpha_\infty$, (20) ensures a linear convergence rate of $\mathbb{E}\left[F(\bar{\mathbf{x}}^T) - f_\star + \|\mathbf{x}^T\|_{\mathbf{K}}^2\right] = \mathcal{O}((1-\delta)^T)$, which shows that the performance of FSPDA-SA is on par with [Nedic et al., 2017, Xu et al., 2017], despite it only requires one round of (sparsified) transmission per iteration.

**Convergence of FSPDA-STORM.** To exploit the benefits of control variates, we need an additional assumption on the stochastic gradient map:

**Assumption 3.8.** *Each stochastic function $f_i(\cdot; \xi)$ is $L_{\mathsf{s}}$-smooth in expectation, i.e., for $i = 1, \ldots, n$,*

$$\mathbb{E}_\xi\left[\|\nabla f_i(\mathbf{x}; \xi) - \nabla f_i(\mathbf{y}; \xi)\|^2\right] \leq L_{\mathsf{s}}^2 \|\mathbf{x} - \mathbf{y}\|^2 \; \forall \, \mathbf{x}, \mathbf{y} \in \mathbb{R}^d. \tag{22}$$

The above assumption is also known as the mean-square smoothness condition, see [Cutkosky and Orabona, 2019], which is strictly stronger than Assumption 3.1. We observe the following convergence guarantee for FSPDA-STORM, whose proof can be found in Appendix D.

---

**Theorem 3.9.** *Under Assumptions 3.1, 3.2, 3.3, 3.4, 3.8. Suppose that the step sizes satisfy the conditions in (184) - (214). Then, for any $T \geq 1$ with the random stopping iteration $\mathsf{T} \sim \mathrm{Unif}\{0, ..., T-1\}$, the iterates generated by FSPDA-STORM satisfy*

$$\mathbb{E}\left[\|\nabla F(\bar{\mathbf{x}}^{\mathsf{T}})\|^2\right] \leq \frac{F_0 - f_\star}{T\alpha/4} + \frac{(\mathtt{e} \cdot 2a_x^2 + \mathtt{f} \cdot 4a_x^2 n)\bar{\sigma}^2}{\alpha/4}, \tag{23}$$

$$\mathbb{E}\left[\sum_{i=1}^n \|\mathbf{x}_i^{\mathsf{T}} - \bar{\mathbf{x}}^{\mathsf{T}}\|^2\right] \leq \frac{F_0 - f_\star}{T\mathtt{a}\gamma\rho_{\min}/8} + \frac{(\mathtt{e} \cdot 2a_x^2 + \mathtt{f} \cdot 4a_x^2 n)\bar{\sigma}^2}{\mathtt{a}\gamma\rho_{\min}/8}, \tag{24}$$

*where the constants $F_0, \mathtt{a}, \mathtt{e}, \mathtt{f}$ are defined in (110).*

---

Setting $\alpha = \mathcal{O}(\bar{\sigma}^{-2/3}T^{-1/3})$, $\eta = \mathcal{O}(n)$, $\gamma = \mathcal{O}(T^{-1/3})$, $\beta = \mathcal{O}(n^{-1}T^{-2/3})$, $a_x = \mathcal{O}(\bar{\sigma}^{-4/3}T^{-2/3})$, $a_\lambda = \mathcal{O}(T^{-1/3})$, $\mathtt{f} = \mathcal{O}(n^{-1}T^{1/3})$ (see (111) - (117)), and initializing the algorithm such that $\|\mathbf{v}^0\|_{\mathbf{K}}^2 = \mathcal{O}(T^{-2/3})$, $\|\overline{\mathbf{m}}_x^0 - (1/n)\mathbf{1}_\otimes^\top \nabla \mathbf{f}(\mathbf{x}^0)\|^2 = \mathcal{O}(T^{-1/3})$ and $\|\mathbf{m}_x^0 - \nabla_{\mathbf{x}}\mathcal{L}(\mathbf{x}^0, \boldsymbol{\lambda}^0)\|^2 = \mathcal{O}(T^{-1/3})$, we have

$$\mathbb{E}\left[\|\nabla F(\bar{\mathbf{x}}^{\mathsf{T}})\|^2\right] = \mathcal{O}\left(\bar{\sigma}^{2/3}/T^{2/3}\right). \tag{25}$$

In regard to the order of $\bar{\sigma}$ and $T$, provided that $n$ is small, the convergence rate of FSPDA-STORM matches the lower bound [Arjevani et al., 2023] for non-convex functions under the same smoothness assumption. Moreover, by the same choice of step sizes, the consensus error converges at the rate of $\mathbb{E}\left[\sum_{i=1}^n \|\mathbf{x}_i^{\mathsf{T}} - \bar{\mathbf{x}}^{\mathsf{T}}\|^2\right] = \mathcal{O}(\bar{\sigma}^{2/3}n\rho_{\min}^{-1}T^{-2/3})$. We remark that in (25), the rate remains constant as $n$ increases such that FSPDA-STORM does not offer the same *linear speedup* observed in Theorem 3.5 for FSPDA-SA. Nevertheless, as $T \gg 1$, the rate of FSPDA-STORM will surpass that of FSPDA-SA and other decentralized algorithms on time varying topologies.

Lastly, we provide detailed discussions on the convergence rates above, e.g., transient time, effects of random topology, etc., in Appendix B.

## 3.1 Insight from Analysis: Fixed Point Iteration of FSPDA-SA

From (8a), the following recursive relationship holds for $\bar{\mathbf{x}}^t$: using the relation $\mathbf{1}^\top \mathbf{A}^\top = \mathbf{0}$, we have

$$\bar{\mathbf{x}}^{t+1} = \bar{\mathbf{x}}^t - \frac{\alpha}{n}\sum_{i=1}^n \nabla f_i(\mathbf{x}_i^t; \xi_i^t). \tag{26}$$

This shows that the evolution of $\{\bar{\mathbf{x}}^t\}_{t\geq 0}$ is similar to that of 'centralized' SGD applied on (1) except that the local gradients are evaluated on the local iterates. However, it is still not straightforward to analyze the convergence of FSPDA-SA as the update of $\mathbf{x}^t$ involves the dual variable $\boldsymbol{\lambda}^t$ which lacks an intuitive interpretation for constructing the right Lyapunov function.

To this end, we study the fixed point(s) of (8) to gain insights. Suppose that for some $t_\star$, the fixed point conditions $\mathbb{E}[\boldsymbol{\lambda}^{t_\star+1} \mid \xi^{:t_\star}] = \boldsymbol{\lambda}^{t_\star}$, $\mathbb{E}[\mathbf{x}^{t_\star+1} \mid \xi^{:t_\star}] = \mathbf{x}^{t_\star}$ hold. Since $\mathbf{R}$ is a diagonal matrix with positive diagonal elements, we observe

$$\mathbb{E}[\boldsymbol{\lambda}^{t_\star+1} \mid \xi^{:t_\star}] = \boldsymbol{\lambda}^{t_\star} \iff \mathbf{R}\mathbf{A}\mathbf{x}^{t_\star} = \mathbf{0} \iff \mathbf{A}\mathbf{x}^{t_\star} = \mathbf{0}, \tag{27}$$

246 On the other hand, the primal update yields

$$\mathbb{E}[\mathbf{x}^{t_\star+1} \mid \xi^{:t_\star}] = \mathbf{x}^{t_\star} - \alpha \nabla \mathbf{f}(\mathbf{x}^{t_\star}) - \eta \mathbf{A}^\top \boldsymbol{\lambda}^{t_\star}. \tag{28}$$

247 Since $\mathbf{x}_1^{t_\star} = \mathbf{x}_2^{t_\star} = \cdots = \mathbf{x}_n^{t_\star}$ at the fixed point (due to (27)), by the consensus condition across two
248 time steps, it implies

$$\begin{aligned}
\mathbb{E}[\mathbf{x}^{t_\star+1} \mid \xi^{:t_\star}] - \mathbf{x}^{t_\star} &= (\mathbf{1} \otimes \mathbf{I}_d)(\bar{\mathbf{x}}^{t_\star+1} - \bar{\mathbf{x}}^{t_\star}) \\
\iff \alpha \nabla \mathbf{f}(\mathbf{x}^{t_\star}) + \eta \mathbf{A}^\top \boldsymbol{\lambda}^{t_\star} &= \tfrac{\alpha}{n}(\mathbf{1}\mathbf{1}^\top \otimes \mathbf{I}_d)\nabla \mathbf{f}(\mathbf{x}^{t_\star}) \\
\iff \eta \mathbf{A}^\top \boldsymbol{\lambda}^{t_\star} &= \alpha \left(\tfrac{1}{n}\mathbf{1}\mathbf{1}^\top - \mathbf{I}_n\right) \otimes \mathbf{I}_d \, \nabla \mathbf{f}((\mathbf{1} \otimes \mathbf{I})\bar{\mathbf{x}}^{t_\star}).
\end{aligned} \tag{29}$$

249 From (29), we see that $\widehat{\boldsymbol{\lambda}}^t$ shall converge to the difference between global and local gradient. Inspired
250 by the above, to facilitate the analysis later, we define

$$\mathbf{v}^t := \mathbf{A}^\top \boldsymbol{\lambda}^t + \tfrac{\alpha}{\eta} \nabla \mathbf{f}((\mathbf{1} \otimes \mathbf{I})\bar{\mathbf{x}}^t), \tag{30}$$

251 for any $t \geq 0$. In particular, we see that $\|\mathbf{v}^t\|_{\mathbf{K}}^2$ measures the violation of (29) in tracking the average
252 deterministic gradient using the dual variables. The latter will be instrumental in analyzing the
253 consensus error bound, as revealed in Lemma C.2.

## 4 Numerical Experiments

255 This section reports the numerical experiments on practical performance of FSPDA. For the time
256 varying topology, we take an extreme setting where for each realization $\mathcal{G}(\xi_a)$, only one edge will
257 be selected uniformly at random from $\mathcal{G}$. We evaluate the performance with the worst-agent metric,
258 i.e., we present the training loss as $\max_{i \in [n]} F(\mathbf{x}_i^t)$, and the stationarity/gradient-norm measure as
259 $\max_{i \in [n]} \|\nabla F(\mathbf{x}_i^t)\|^2$. This captures the worst-case of the solutions produced by the algorithms.
260 Unless otherwise specified, all algorithms are initialized with $\mathbf{x}_i^0 = \bar{\mathbf{x}}^0$, and for FSPDA we initialize
261 $\widehat{\boldsymbol{\lambda}}^0 = \mathbf{m}_{x,i}^0 = \mathbf{m}_{\lambda,i}^0 = \mathbf{0}$, and the stochastic gradients are estimated with a batch size of 256. In the
262 interest of space, omitted details and hyperparameters of the experiments can be found in Appendix F.

263 **MNIST Experiments.** The first set of experiments considers a moderate-scale setting of training a
264 one hidden layer feed-forward neural network with 100 hidden neurons (total number of parameters
265 $d = 79{,}510$) on the MNIST dataset with $m = 60{,}000$ samples of 784-dimensional features.

266 In the first experiment, we consider the static topology $\mathcal{G}$ as an Erdos-Renyi graph with connectivity of
267 $p = 0.5$ and $n = 10$ agents. We compare the proposed FSPDA-SA, FSPDA-STORM with six benchmark
268 algorithms utilizing different types of time-varying topology. Among them, DSGD [Koloskova et al.,
269 2020] and Swarm-SGD [Nadiradze et al., 2021] use the general time varying topology setting as FSPDA
270 where each edge of $\mathcal{G}(\xi_a)$ is active uniformly at random, in addition to random sparsification used
271 FSPDA-SA and adaptive quantized used in Swarm-SGD; CHOCO-SGD [Koloskova et al., 2019b] takes
272 $\mathcal{G}(\xi_a)$ as an broadcasting subgraph where one agent selects all his/her neighbors; Decen-Scaffnew
273 [Mishchenko et al., 2022], LED [Alghunaim, 2024], and K-GT [Liu et al., 2024] utilize local updates
274 where $\mathcal{G}(\xi_a)$ is either taken as an empty topology, or as the static topology $\mathcal{G}$. We configure these
275 algorithms such that they have the same communication cost (in terms of bits transmitted over
276 network) *on average*. For instance, the local update algorithms (Decen-Scaffnew, LED, K-GT)
277 only communicate once using $\mathcal{G}$ every $\mathcal{O}\left(\frac{|\mathcal{E}|d}{k}\right)$ iterations to match the communication cost of
278 $k$-coordinate sparse one-edge random graph used in FSPDA.

279 The local objective function held by each agent is the cross-entropy classification loss on a local
280 dataset with $m_i = 6000$ samples, plus a regularization loss $\frac{\lambda}{2}\|\mathbf{x}_i\|^2$ with $\lambda = 10^{-4}$, where $\mathbf{x}_i$ are the
281 weight parameters of the feed-forward neural network classifier. We split the training set into $n = 10$
282 disjoint sets such that each set contains only one class label and assign each set to one agent as its
283 local dataset. Note that as we do not shuffle the data samples across local datasets, the local objective
284 function held by different agents will become highly heterogeneous.

285 Fig. 1 compares the squared gradient norm, training loss, consensus error of the benchmarked algo-
286 rithms. We first note that both FSPDA algorithms have significantly outperformed DSGD, Swarm-SGD
287 on the general time varying topology as well as CHOCO-SGD. Meanwhile, the performance of FSPDA
288 is comparable to the local update algorithms Decen-Scaffnew, LED, K-GT. Notice that the latter

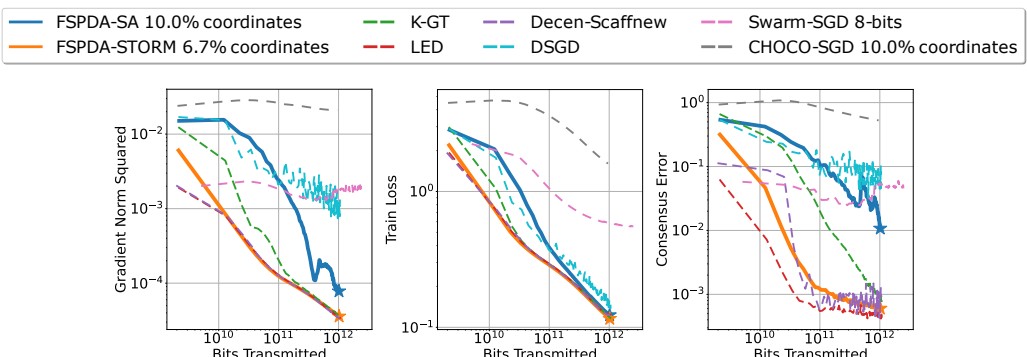

Figure 1: Feed-forward neural network classification training on MNIST using $10^6$ iterations.

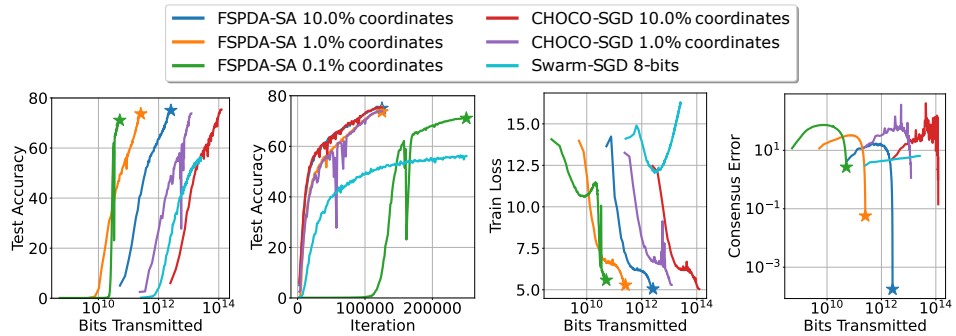

Figure 2: Resnet-50 classification training on Imagenet.

require additional synchronization steps which may not be suitable for random networks. Lastly, we notice that as $T \gg 1$, FSPDA-STORM can slightly outperform FSPDA-SA due to its $\mathcal{O}(1/T^{2/3})$ rate as shown in our analysis. We further expand the experiments by a series of ablation studies over data heterogeneity, sparsity levels, graph topologies, gradient noise and dual momentum in Appendix E.

**Imagenet Experiments.** The second set of experiments consider a large-scale setting for training a Resnet-50 network (total number of parameters $d = 25,557,032$) on the Imagenet dataset (training dataset of 1,281,168 images from 100 classes, re-scaled and cropped to $256 \times 256$ image dimensions). We consider cross-entropy classification loss plus the same L2 norm regularization loss as in the previous setup. We split the dataset across a network of $n = 8$ nodes where the static graph $\mathcal{G}$ is taken as the fully connected topology. The performance metrics are measured at the network average iterate $\bar{\mathbf{x}}^t$. Inspired by [Loshchilov and Hutter, 2016, Eq. (5)] we adopt a cosine learning rate scheduling with 5 epochs of linear warm up for every algorithm. In particular, the step sizes $\alpha, \eta$ of FSPDA-SA are scheduled simultaneously such that $\alpha_t/\eta_t$ remains constant, as illustrated in Appendix F. We draw a batch of 128 samples to estimate the stochastic gradient.

We focus on the communication efficiency and only compare FSPDA-SA, CHOCO-SGD, Swarm-SGD in this experiment due to limited resources. The results are reported in Figure 2 that compare the test accuracy and training loss against iteration number and bits transmitted. When compared with CHOCO-SGD, FSPDA-SA achieves almost the same accuracy using one-edge random graphs with at least 100x reduction in communication cost on 100 epoch training. Also notice that further compressing the communication to 0.1% sparse coordinates in FSPDA-SA requires more training epochs to recover the same level of accuracy.

**Conclusions.** This paper proposed a fully stochastic primal dual gradient algorithm (FSPDA) framework for decentralized optimization over arbitrarily time varying random networks. We utilize a new stochastic augmented Lagrangian function and apply SA to search for its saddle point. We develop two algorithms, one is by plain SA (FSPDA-SA), and one uses control variates for variance reduction (FSPDA-STORM). We prove that both algorithms achieve state-of-the-art convergence rates, while relaxing assumptions on both bounded heterogeneity and the type of time varying topologies.

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
