# OpenReview forum: "A Stochastic Approximation Approach for Efficient Decentralized Optimization on Random Networks"
_NeurIPS.cc/2025/Conference — Submitted to NeurIPS 2025_

### Official Review · Reviewer_Rmsh · 2025-06-30

**Clarity:** 3
**Significance:** 3
**Originality:** 3
**Rating:** 5
**Confidence:** 4

**Summary:**

This submission is concerned with the problem of the distributed optimization over time-varying nondirected communication graphs of a smooth (possibly nonconvex) function given as a sum $F(x) = \frac{1}{n} \sum_{i=1}^{n} f _i(x)$, where and each $f _i : \mathbb{R}^d \mapsto \mathbb{R}$ is exclusively available to agent $i$ though noisy measurements of its gradient ($i=1,\dots,n$). The problem is reformulated as the constrained optimization problem

$F(x) = \frac{1}{n} \sum_{i=1}^{n} f _i(\mathbf{x} _i)$ subject to $\mathbf{x} _1 = \cdots = \mathbf{x} _n$,

and two algorithms are proposed: FSPDA-SA and FSPDA-STORM. Both algorithms use stochastic approximation (SA) (in primal-dual form) to reach a saddle point $\mathbf{x}^\ast = (\mathbf{x}^\ast _1, \dots , \mathbf{x}^\ast _n)$ of a (stochastic) augmented Lagrangian for the constrained problem, which implies finding a stationary point $x^\ast$ of the initial problem with consensus on the individual variables $\mathbf{x}^\ast _1= \cdots = \mathbf{x}^\ast _n = x^\ast$.
The two algorithms can be implemented asynchronously for the agents, and FSPDA-STORM has the specificity that momentum is used in order to reduce the variance due to inhomogeneity in the local objective functions, measurement noise and changing graph topologies. Analysis shows that FSPDA-SA matches existing methods in terms of convergence rate ($O(1/\sqrt{T})$), while offering more flexibility in implementation, and FSPDA-STORM reaches the faster rate $O(1/T^{2/3})$. Numerical experiments are provided for benchmark problems.

**Questions:**

1. I was confused by Assumptions 3.3 and 3.4. As the assumptions are stated, it looks like one can find constants $\sigma _i(\mathbf{x} _i)$ and $\sigma _A(\mathbf{x})$ for every new $\mathbf{x} _i$ and $\mathbf{x}$. Are these not uniform bounds for all $\mathbf{x} _i$, $\mathbf{x}$ (as one would imagine and as apparent from the developments)? One way to reformulate the assumptions then would for instance be "For $i\in [n]$, there exists $\sigma _I \geq 0$ such that $\dots$ holds for all $\mathbf{x} _i \in \mathbb{R}^d$ $\dots$". Could the authors please clarify the statements?

2. In (6), (8b), I believe there is a sign issue and the variables $\mathbf{x} _j$ and $\mathbf{x} _i$ should be permutated. Could the authors please double-check these equations?

3. In (10c) and (10d), I think three $\pm$ signs should be $\mp$ here too.

4. In Corollary 3.7, it would be useful to give the expression for $F _t$ (even with unspecified parameters), which I believe is only available in the appendix. And in (20), what is $\mathbb{E} _t$? Is it the same conditional expectation as $\mathbb{E} [ \cdot | \xi^{:t}]$ in Section 3.1? I do not remember these notations being introduced  anywhere in the paper.

5. On Line 211, how did you choose the value for the stepsize $\alpha$ in (20)? I was unable to reproduce its derivation based on Theorems 3.5 and C.5. And what is $c$ on Line 212? Is is the same parameter as $\mathbf{c}$ in (45)?

6. In Algorithm 3 (asynchronous implementation), I understood $\hat{c} _i$ as an estimate of the relative frequency of gradient updates at node $i$. Hence my intuition was to find the factors $1/\hat{c} _i$ (in place of $\hat{c} _i$) in (32) and (35). Could the authors please explain the $\hat{c} _i$ factors in the two equations? Also, does this technique rest on the assumption of stationary update frequencies for all agents?

7. In (58), I got lost at the time of deriving the last two terms. The reader might welcome a word of explanation here. Did you use Hölder's inequality and the property that $\mathbf{K}\mathbf{K}=\mathbf{K}$?

Minor comments:

On Line 80: time varying topology -> time-varying topology

On Line 107: $\mathcal{N} _i(\xi)$ the neighborhood of the $i$th agent as -> $\mathcal{N} _i(\xi)$, the neighborhood of the $i$th agent, as

In (29), (30): $I$ -> $I_d$

On Line 750: The neighbor $j$ whose communicated with -> The neighbor $j$ who has communicated with

**Ethical Concerns:**

["NO or VERY MINOR ethics concerns only"]

**Final Justification:**

The proposed FSPDA algorithm is an extension to time-varying networks of the EXTRA algorithm for nonconvex optimization, where the linear constraint enforcing consensus from which the augmented Lagrangian is derived now has the form of an expectation. The specific requirements of the algorithm are that the graph must be connected only in expectation (not at each iteration), and no bound is needed on the heterogeneity of the local objective functions.

In my sense, the proposed algorithm and the various techniques that are used are not particularly innovative, but the analysis looks technically sound to me, and it is original for nonconvex optimization in the considered time-varying setting (nondirected time-varying graphs that are only connected in expectation) and without more stringent assumptions. In particular, no bounds on heterogeneity are required in the analysis, which is a valuable feature of FSPDA, compared to the methods that rely on such bounds and tend to slow down in the case of highly heterogeneous local objective functions (as apparent from the numerical experiments in Appendix E.1).

**Limitations:**

yes

**Paper Formatting Concerns:**

I did not detect formatting issues.

**Quality:**

3

**Strengths And Weaknesses:**

I thought the submission was interesting and well written. In my view (and in the light of Table 1), the technical contributions are significant. The discussions are also rich and useful (model and assumptions, interpretation of the results, comparison with other methods, proof sketches, etc.), which makes the paper accessible to non-experts on the topic. The overall organization of the paper is satisfying, as the main text summarizes well the many pages of computations that were relegated to the appendices. A strong point point of the submission is the extensive tests on several benchmark problems and from many viewpoints (convergence, communication costs, impact of noise, graph sparsity, asynchrony, the acceleration technique, etc.).

I was not able to proofread all the derivations in equal detail, but the appendices I did review were carefully written and technically sound. Most of the developments rely on common concepts from graph theory and function analysis, and on inequalities and techniques typically used in convergence analysis.

My comments below are mostly concerned with the presentation. I saw a few typos but no real weakness in the submission.

---

> ### Author Rebuttal · Authors · 2025-07-30
>
> Thank you for your careful review for our paper and constructive comments. Please find our point-to-point responses below.
>
> > 1. I was confused by Assumptions 3.3 and 3.4. As the assumptions are stated, it looks like one can find constants $\sigma_i(\mathbf{x}_i)$ and $\sigma_A(\mathbf{x})$ for every new $\mathbf{x}_i$ and $\mathbf{x}$. Are these not uniform bounds for all $\mathbf{x}_i$, $\mathbf{x}$ (as one would imagine and as apparent from the developments)? One way to reformulate the assumptions then would for instance be "For $i\in [n]$, there exists $\sigma_I \ge 0$ such that ... holds for all $\mathbf{x}_i \in \mathbb{R}^d$". Could the authors please clarify the statements?
>
> The statements were indeed confusing. We will update the statements according to your suggestions.
>
> > 2. In (6), (8b), I believe there is a sign issue and the variables $\mathbf{x}_j$ and $\mathbf{x}_i$ should be permutated. Could the authors please double-check these equations?
>
> > 3. In (10c) and (10d), I think three $\pm$ signs should be $\mp$ here too.
>
> We apologize for the typos in (6), where the correct sign should be permuted as $(\mathbf{x}_i - \mathbf{x}_j)$. Similarly, (8) should be $(\mathbf{x}_i - \mathbf{x}_j)$, (10c) is correct in the submission, (10d) should be $(\mathbf{x}_i - \mathbf{x}_j)$. We will implement these corrections in the revision.
>
>
> > 4. In Corollary 3.7, it would be useful to give the expression for $F_t$ (even with unspecified parameters), which I believe is only available in the appendix. And in (20), what is $\mathbb{E}_t$? Is it the same conditional expectation as $\mathbb{E}[\cdot | \xi^{:t}]$ in Section 3.1? I do not remember these notations being introduced anywhere in the paper.
>
> Thank you for the suggestion, we will include the expression for $F_t$ when extra space is allotted in the camera-ready version. $\mathbb{E}_t$ in (20) **is the conditional expectation** conditioned on $\xi^t$.
>
> > 5. On Line 211, how did you choose the value for the stepsize in (20)? I was unable to reproduce its derivation based on Theorems 3.5 and C.5. And what is on Line 212? Is is the same parameter as $\mathbf{c}$ in (45)?
>
> The choice of $\alpha$ in (21) is due to the result of Appendix C.6, where we can telescope the inequality (83) and upper bound the geometric sum of $\sum_{k=0}^{t}(1-\delta)^{k} \leq 1/(1-(1-\delta)) = 1/\delta$ where $\delta = O(\alpha)$ due to the condition in line 886. We then obtain $F_{T} - f_\star = O((1-\delta)^T(F_0 - f_\star) + \mathbb{C}_{\sigma} \alpha^2 \bar{\sigma}^2 / \delta)$ and $\delta = O(\alpha) = O(\ln(T)/(n^2T))$ is the optimal choice for the above bound.
>
> The constant $c$ represents the omitted constants in the above big-$\mathcal{O}$ notations and is different from $\mathtt{c}$ in (45). For clarity, we will rewrite line 211 as "By setting $\alpha = \Theta( \ln(T) / (n^2 T))$ in (20), ..." in the revision.
>
> > 6. In Algorithm 3 (asynchronous implementation), I understood $\hat{c}_i$ as an estimate of the relative frequency of gradient updates at node $i$. Hence my intuition was to find the factors $1/\hat{c}_i$ (in place of $\hat{c}_i$) in (32) and (35). Could the authors please explain the $\hat{c}_i$ factors in the two equations? Also, does this technique rest on the assumption of stationary update frequencies for all agents?
>
> Thank you for the careful observation. You are right that the factors regarding $\hat{c}_i$ should be inversed. $\hat{c}_i$ is introduced to maintain the unbiasedness of stochastic gradient. We will correct the typo by rewrting (34) as $\hat{c}_i = (t_i' + 1) / g_i$ if $\nabla f_i(\cdots)$ is ready. The same applies to (32). This unbiased scaling does depend on the assumption that the update frequency is stationary, i.e., $\hat{c}_i \sim \mathcal{D}\_{c_i}$ for a fixed distribution $\mathcal{D}\_{c_i}$ for the sake of analysis.
>
>
> > 7. In (58), I got lost at the time of deriving the last two terms. The reader might welcome a word of explanation here. Did you use Hölder's inequality and the property that $\mathbf{K}\mathbf{K}=\mathbf{K}$?
>
> You are right. The last two terms in (58) are derived from applying Young's inequality and using the fact ${\bf K}^\top {\bf K} = {\bf K} {\bf K} = {\bf K}$. Particularly, for any $\alpha>0$, we observe that
>
> $2 \braket{(\mathbf{I} - \gamma \mathbf{L})\mathbf{x}^t| \mathbf{e}^t_g}\_{{\bf K}}$
>
> $= 2 \braket{{\bf K} (\mathbf{I} - \gamma \mathbf{L})\mathbf{x}^t| {\bf K} \mathbf{e}^t_g}$
>
> $\leq \alpha || (\mathbf{I} - \gamma \mathbf{L})\mathbf{x}^t ||\_{\bf K}^2 + \frac{1}{\alpha} || \mathbf{e}^t_g ||\_{\bf K}^2$
>
> > A list of typos suggested by the reviewer.
>
> We will correct them in the revision. Thank you.

---

> > ### Comment · Reviewer_Rmsh · 2025-08-06
> >
> > Thank you very much for your detailed feedback. You have addressed all my questions and comments.

---

### Official Review · Reviewer_TNhj · 2025-06-30

**Clarity:** 2
**Significance:** 3
**Originality:** 3
**Rating:** 3
**Confidence:** 4

**Summary:**

This paper proposes the FSPDA framework for decentralized optimization on random networks. While tackling time-varying topologies is relevant, the work suffers from fundamental flaws in novelty, technical rigor, and validation that preclude publication. The claims of outperforming existing methods are unsupported and theoretical analyses contain critical gaps.

**Questions:**

1.The paper positions FSPDA as the "first" solution for random topologies, but prior works (Kovalev et al., 2021; Li & Lin, 2024) explicitly handle time-varying graphs. The distinction between gradient tracking (Qu & Li, 2017) and primal-dual methods (Hajinezhad & Hong, 2019) is superficial.

2.Assumption 3.4 Unverifiable: The bound $\sigma_A^2 \propto\|x\|_K^2$ (Eq. 15) is non-standard and lacks proof. For sparse graphs, $\sigma_A$ likely scales with $1 / \sqrt{|\mathrm{E}|}$, contradicting the analysis.

3.Parameter Infeasibility: Conditions (46) for Theorem 3.5 require $\delta_1 \geq 8$ and $\gamma_{\infty} \sim \mathrm{O}\left(\sigma_A^{-2}\right)$. When $\sigma_A$ is large (e.g., highly sparse graphs), $\gamma_{\infty}$ becomes vanishingly small, stalling convergence. No feasible configuration is demonstrated.

4.Omitted Dependencies: The $\mathrm{O}(1 / \sqrt{n T})$ rate (Eq. 19) misleadingly hides $\rho_{\min }^{-1}$ and $\sigma_A^4$ terms (Eq. 204), voiding linear speedup guarantees.

5. The emphasis on "fully stochastic" networks ignores that most real-world systems (e.g., federated learning) exhibit structured time-variation (e.g., periodic updates), handled better by local-GT.

**Ethical Concerns:**

["NO or VERY MINOR ethics concerns only"]

**Final Justification:**

Thank you very much for your detailed feedback. You have answered all my questions and opinions. I will improve my score.

**Limitations:**

- Appendix A notes that asynchronous FSPDA has "suboptimal" convergence but provides no quantitative analysis (e.g., convergence rate degradation under delays).
- §3.2 mentions $\sigma_A$ (topology noise) without examining real-world feasibility (e.g., how $\sigma_A$ scales with edge sparsity $p$ in dynamic networks).
- Local updates are called "suboptimal" (§1.3) but lack comparison to SOTA (e.g., ProxSkip).

**Paper Formatting Concerns:**

I didn't find any major formatting issues in this article.

**Quality:**

3

**Strengths And Weaknesses:**

Strengths
- Effectively addresses the critical gap in decentralized optimization for random time-varying topologies (§1.5), which is highly relevant to real-world dynamic networks (e.g., UAV swarms, edge computing).
- Proposes a unified stochastic augmented Lagrangian framework (Eq. 4) that concurrently models:
- Data randomness $\left(\xi_i\right)$
- Topology randomness $\left(\xi_a\right)$.

Weaknesses
- $v^t$ variable inconsistency: Definition $v^t=A^{\top} \lambda^t+\frac{\alpha}{\eta} \nabla f\left(1_{\otimes} \bar{x}^t\right)$ (Eq. 30) requires global gradient knowledge, violating decentralization principles.
- Unverifiable assumptions: $\sigma_A^2 \propto\|x\|_K^2$ (Eq. 15) lacks empirical/theoretical justification and fails for sparse graphs.
- Infeasible parameter conditions: Step-size constraints (Eq. 46) become unsolvable when $\sigma_A>1$ (common in sparse networks).
- Ambiguous convergence claims: $O(1 / \sqrt{n T})$ rate (Eq. 19) hides $\sigma_A^4$ dominance (p. 204)
- Critical notation overload: $\mathrm{L}, v^t, C_{i j}(\xi)$ redefined without consolidation

---

> ### Author Rebuttal · Authors · 2025-07-30
>
> Thank you for the critical and constructive comments. We believe there are some misunderstandings by the reviewer, and wish to clarify them in the point-to-point responses below.
>
> > 1. The paper positions FSPDA as the "first" solution for random topologies, but prior works (Kovalev et al., 2021; Li & Lin, 2024) explicitly handle time-varying graphs. The distinction between gradient tracking (Qu & Li, 2017) and primal-dual methods (Hajinezhad & Hong, 2019) is superficial.
>
> The novelty of FSPDA lies on being a "fast" decentralized algorithm for time-varying (TV) graphs under the stochastic non-convex optimization setting - as illustrated in Table 1 - note that by "fast" algorithm, we refer to ones that converge without restrictions such as bounded heterogeneity. As hinted by prior works on static graphs, such "fast" algorithm design can only be achieved using gradient tracking (GT) or primal-dual (PD) methods, note GT can be viewed as a special case of PD [Chang et al., 2020].
>
> The analysis in [Kovalev et al., 2021, Li & Lin, 2024] (and [Nedic et al., 2017]) are applicable to GT on TV graphs, but are limited to strongly convex functions under the **deterministic** gradient setting. These are fundamentally different settings from our non-convex & stochastic cases as the iterates are expected to converge to a unique solution. Indeed, it was explicitly pointed out in [Koloskova et al., 2021] that analyzing the latter case with GT is highly non-trivial (in the non-convex setting) as it involves "*the non-symmetric product of two mixing matrices*", and therefore it only analyzed the simpler DSGD algorithm whose convergence are restricted by the bounded heterogeneity condition.
>
>
> > 2. Assumption 3.4 Unverifiable: The bound $\sigma_A^2 \propto \|x\|_{K}^2$ (Eq. 15) is non-standard and lacks proof. For sparse graphs, $\sigma_A$ likely scales with $1 / \sqrt{|E|}$, contradicting the analysis.
>
> > §3.2 mentions $\sigma_A$ (topology noise) without examining real-world feasibility (e.g., how $\sigma_A$ scales with edge sparsity $p$ in dynamic networks).
>
> We should have highlighted Assumption 3.4 as one of our technical novelty to overcome the challenges in analyzing "fast" algorithm over TV graphs - as the latter is applied in Lemma C.2 for controlling the consensus error recursion. As an illustration, the assumption can be verified easily by the crude bound below:
>
> $\mathbb{E}[|| \mathbf{A}^\top \mathbf{A}(\xi_a) \mathbf{x} - \mathbf{A}^\top \mathbf{R} \mathbf{A} \mathbf{x} ||^2]$
>
> $=\mathbb{E}[|| \mathbf{A}^\top {\bf I}\_{nd}(\xi_a) \mathbf{A} \mathbf{x} - \mathbf{A}^\top \mathbf{R} \mathbf{A} \mathbf{x} ||^2]$
>
> $= \mathbb{E}[|| (\mathbf{A}^\top {\bf I}\_{nd}(\xi_a) - \mathbf{A}^\top \mathbf{R} ) ( \mathbf{A} \mathbf{x}) ||^2]$
>
> $\leq \mathbb{E}[ || \mathbf{A}^\top {\bf I}\_{nd}(\xi_a)- \mathbf{A}^\top \mathbf{R} ||^2 ] \cdot || \mathbf{A} \mathbf{x} ||^2$
>
> $\leq \mathbb{E}[ || {\bf I}\_{nd}(\xi_a)- \mathbf{R} ||^2 ] \cdot || {\bf A} ||^2 \cdot \bar{\rho}_{\max} \cdot || \mathbf{x} ||^2\_{\mathbf{K}}$
>
> where $\mathbf{R} = \mathbb{E}[{\bf I}_{nd}(\xi_a)]$ and the last inequality is due to Assumption 3.2. This shows that
>
> $\sigma_A^2 \leq \mathbb{E}[ || {\bf I}\_{nd}(\xi_a)- \mathbf{R} ||^2 ] \cdot \bar{\rho}_{\max} \cdot || {\bf A} ||^2$.
>
> Note the above bound can be improved. In the case when a $p$-sparse graph are used for the communication steps of FSPDA that selects an edge from $\mathcal{G}$ with probability $p$, we have $\mathbb{E}[ || {\bf I}\_{nd}(\xi_a)- \mathbf{R} ||^2 ] \leq (1-p)^2$ and thus $\sigma_A^2 = (1-p)^2 \bar{\rho}_{\max} || {\bf A} ||^2$. Note one has $|| {\bf A} ||^2 \leq O( \mathcal{E} )$.
>
> The reviewer is right that $\sigma_A^2$ is likely to scale with $|\mathcal{E}|$ especially in the extreme case of $p = 1/|\mathcal{E}|$. However, we politely disagree that it contradicts with our analysis. In practice, the graph size is finite and so is $|\mathcal{E}|$, and it remains feasible to find a stepsize configuration attaining fast convergence rate for FSPDA.
>
>
>
>
> > 3. Parameter Infeasibility: Conditions (46) for Theorem 3.5 require $\delta_1 \ge 8$ and $\gamma_\infty \sim \mathcal{O}(\sigma_A^{-2})$. When $\sigma_A$ is large (e.g., highly sparse graphs), $\gamma_{\infty}$ becomes vanishingly small, stalling convergence. No feasible configuration is demonstrated.
>
> It remains feasible to satisfy (46) by picking $\alpha = \Theta(1/\sqrt{T})$ for large $T$. A possible choice is to have $\gamma = \gamma_{\infty} = O(\rho_{\min}/\sigma_A^2)$, $\eta = \eta_{\infty} = O(\rho_{\min}^3/\sigma_A^2)$. With $\alpha = \Theta(1/\sqrt{T})$ such that for large enough $T$, it satisfies $\alpha \leq \alpha_{\infty} = O(\rho_{\min}^6/\sigma_A^4)$. The above configuration is feasible, as long as $\sigma_A < \infty$ is finite and **independent** of $T$, where they guarantee the convergence of FSPDA-SA.
>
> We understand the concern that the above choices may lead to small parameters when random sparse graph sampled from $\mathcal{G}$ with large $|\mathcal{E}|$ are used in FSPDA. Besides that they remain feasible for finite $|\mathcal{E}|$, we wish to point out that the effects of $\sigma_A, \rho_{\min}$ vanishes as $T \gg 1$ (also see the clarification to Q4 below). It can be seen in Theorem C.5 (full ver of Theorem 3.5), where one has $\texttt{a} = O(1/\sqrt{T}), \alpha = O(1/\sqrt{T})$ and it will make the last terms in (50) vanish.
>
> Additionally, we notice that such step size choices are common for the TV graph setting. For example, in Theorem 1 of [Li & Lin, 2024], the step size for GT is $O(1/B^4)$ for $B$-connected TV graph. With extremely sparse TV graph, one has $B \propto |\mathcal{E}|$, their analysis also suggests a vanishingly small step size parameter.
>
> Lastly, our experiment results demonstrate extremely sparsified cases where the random graphs are drawn as one-edge graphs in Fig 1, 2. In Fig 5 of Appendix E, different degrees of sparsification are also demonstrated. All of our experiment results demonstrate convergence in practical neural network training problems. The hyperparameter configurations of every experiments are provided in Appendix F.
>
>
> > 4. Omitted Dependencies: The $\mathcal{O}(1/\sqrt{nT}$ rate (Eq. 19) misleadingly hides $\rho_{\min}^{-1}$ and $\sigma_A^4$ terms (Eq. 204), voiding linear speedup guarantees.
>
> We believe that the reviewer referred to L204 on p.6. The sentence actually discusses the convergence rate when the gradients used in FSPDA-SA are deterministic with $\bar{\sigma} = 0$, where it is irrelevant to the linear speedup rate in (19) as the latter is discussed for cases with stochastic gradient $\bar{\sigma} > 0$. As discussed in [Lian et al., 2017], such linear speedup behavior (such that the decentralized algorithm matches the performance of a centralized one) is only relevant when stochastic gradients are used.
>
>
> > 5. The emphasis on "fully stochastic" networks ignores that most real-world systems (e.g., federated learning) exhibit structured time-variation (e.g., periodic updates), handled better by local-GT.
>
> > Local updates are called "suboptimal" (§1.3) but lack comparison to SOTA (e.g., ProxSkip).
>
> In contrary, it is a unique advantage of FSPDA to handle fully stochastic networks with unstructured time variation while performing local updates asynchronously - see Algo. 2 & 3 - note Algo. 2 & 3 also includes structured time variation update as special cases. In addition to offering strictly better flexibility, the achieved convergence rate remains comparable to those with specialized analysis in the dominant $O(1/\sqrt{nT})$ term. We admit that our analysis, which is obtained without structured TV graph, maybe less tight than prior works such as [R1] and is thus said to be "suboptimal" - We apologize for the unclear wordings. Yet we found that the differences in rate mostly appear in the transient terms that vanish as $T \gg 1$, see Theorem C.5 & Theorem D.8.
>
> Lastly, we refer the reviewer to the experiments in Fig 1. Here, we show that using an unstructured time variation updates, FSPDA is able to achieve comparable performance, or outperform algorithms utilizing specialized structured TV / periodic updates, e.g., DecenScaffnew/ProxSkip in [Mishchenko et al., 2022] & [R1].
>
> [R1] L. Guo, S.A. Alghunaim, K. Yuan, L. Condat, J. Cao, Achieving Linear Speedup with ProxSkip in Distributed Stochastic Optimization, arxiv/2310.07983.
>
>
> > Appendix A notes that asynchronous FSPDA has "suboptimal" convergence but provides no quantitative analysis (e.g., convergence rate degradation under delays).
>
> For the convergence rate under delays with stochastic gradients, it can be quantified as having an increased gradient variance $\bar{\sigma}$ as discussed in Section 2.1. Similar to how we handle random graphs, our model of asynchrony is only in concern with the rate of successful access to stochastic gradient $\bar{b}_i$ and the rate of edge activation $\sigma_A, \rho\_{\min}$, rather than the more restrictive deterministic delay bound that is commonly adopted in the existing asynchronous algorithms. In particular, by Algorithm 1 in Appendix A, each agent always apply the latest stochastic gradient (see (32), (35)), instead of stale gradient from past iterates. This distinguish our asynchronous algorithm from the existing delay-bounded algorithms that utilizes stale gradient updates.
>
> We look forward to discussing with the reviewer if any further clarifications are needed.

---

> > ### Author Response · Authors · 2025-08-07
> >
> > Dear Reviewer TNhj,
> >
> > As the end of discussion period is approaching, we would like to see if our rebuttal has addressed your concern. We kindly remind the reviewer that you can follow up with new questions to our rebuttal, or raise the rating of our submission if you are satisfied with our response.
> >
> > Best regards,
> >
> > Authors of Paper 26853.

---

> > ### Comment · Reviewer_TNhj · 2025-08-09
> >
> > Thank you very much for your detailed feedback. You have answered all my questions and opinions. I will improve my score.

---

### Official Review · Reviewer_RPmm · 2025-07-02

**Clarity:** 4
**Significance:** 4
**Originality:** 4
**Rating:** 6
**Confidence:** 5

**Summary:**

The authors present a stochastic, primal-dual approximation algorithm that can be used with random, time-varying networks under unreliable communication bandwidth.  The randomness in time-varying networks is formulated as a stochastic augmented Lagrangian function whose expectation yields saddle points that coincide with stationary solutions of the decentralized optimization. Two variants of the proposed algorithms developed from this framework are (1)  FSPDA -- which allows agents to execute multiple local gradient steps depending on the time-varying topology, and (2) FSPDA-STORM, which incorporates variance reduction steps to improve sample complexity. Empirical results are presented on the MNIST and ImageNet datasets using an extreme setting for the realization of the time-varying topology; baselines are presented with SOTA algorithms such as CHOCO SGD, Swarm-SGD and others.

**Questions:**

Empirical analysis is performed on small, complete random graph topologies. What is the relationship between the dynamic nature of the graph topology, the size of the graph, the communication cost incurred on it,  and local computation time? Do larger graphs experience bottlenecks that affect eventual convergence?

**Ethical Concerns:**

["NO or VERY MINOR ethics concerns only"]

**Final Justification:**

I have read the rebuttal of the authors, including extra empirical results presented. I have also carefully read the discussions and think my score remains justified.

**Quality:**

4

**Strengths And Weaknesses:**

The paper is very well written and addresses a novel problem -- that of achieving fast convergence on time-varying random networks. Prior related work in the domain of interest has been discussed clearly.

The problem is formulated as a stochastic augmented Lagrangian function; however, extension to time-varying random networks is not straightforward due to inconsistency of dual variables. This challenge is overcome by proposing a stochastic equality constrained reformulation, and two related algorithms FSPDA and FSPDA-STORM are presented.

The implementation of 8a and 8b in Appendix A is quite important to understand the flow of the algorithm FSPDA. Perhaps it would be useful to consider having this description in the main paper at the cost of the some convergence analysis proofs.

Empirical analysis -- while the extreme setting of only one edge being selected at random is interesting, the effect of the convergence of the algorithm on larger random networks would need to be studied. For example, in MNIST, the number of agents = 10, and in ResNet, it is 8. These are reasonably small to assess the overall performance of the algorithm. For instance, what is the relationship between the dynamic nature of the graph topology, the size of the graph, the communication cost incurred on it,  and local computation time? Do larger graphs experience bottlenecks that affect eventual convergence?

---

> ### Author Rebuttal · Authors · 2025-07-31
>
> Thank you for your careful review for our paper and constructive comments. Please find our point-to-point responses below.
>
> > The implementation of 8a and 8b in Appendix A is quite important to understand the flow of the algorithm FSPDA. Perhaps it would be useful to consider having this description in the main paper at the cost of the some convergence analysis proofs.
>
> Subject to page limitation, we will try to incorporate the asynchronous implementation in the main text to provide a self-contained discussions in the paper.
>
> > Empirical analysis -- while the extreme setting of only one edge being selected at random is interesting, the effect of the convergence of the algorithm on larger random networks would need to be studied. For example, in MNIST, the number of agents = 10, and in ResNet, it is 8. These are reasonably small to assess the overall performance of the algorithm. For instance, what is the relationship between the dynamic nature of the graph topology, the size of the graph, the communication cost incurred on it, and local computation time? Do larger graphs experience bottlenecks that affect eventual convergence?
>
> > Empirical analysis is performed on small, complete random graph topologies. What is the relationship between the dynamic nature of the graph topology, the size of the graph, the communication cost incurred on it, and local computation time? Do larger graphs experience bottlenecks that affect eventual convergence?
>
> Thanks for the suggestions. Due to limited time and compute resources, we are only able to extend the experiments under the setting of Fig. 1 to test the effect of graph sizes on convergence, and verifying the linear speedup effects in (19). Instead of complete graphs, we also use ER graphs (with connectivity p=0.5) in these extended experiments.
>
> From our theory, larger graphs actually tend to improve convergence due to the linear speedup effects as shown in (19), albeit such effect may only become dominant as $t \gg 1$. To test it, we fix the batch size of each agent so that the local computation time is equivalent across different graph size. We further fix the communication cost of each edge across $n=10,20,40$ by keeping the edge selection probabilities constant. To verify the linear speedup effects, we terminate the FSPDA-SA algorithm after the same number of training samples across agents are used:
>
> | Iteration | t=1.25e6 | t=2.5e6 | t=5e6   | t=1e7   |
> |-----------|----------|---------|---------|---------|
> | $n=10$      | 5.01e-5  | 2.91e-5 | 2.61e-5 | **2.58e-5** |
> | $n=20$      | 6.74e-5  | 2.88e-5 |    **2.37e-5**    |         |
> | $n=40$      | 4.24e-5  | **3.58e-5** |         |         |
>
> *Table A. Ablation study on the effect of graph size $n$. Each run uses 1% coordinate sparsity and random graphs of edge probability 0.0222 $(n=20, 40)$ or 1-edge random graphs $(n=10)$. The table records $\max_{i\in[n]} ||\nabla F(\mathbf{x}_i^t) ||^2$.*
>
> We observe that similar solution stationarity is achieved for $nT = 1e8$ (i.e., the highlighted entries) regardless of $n$. This confirms with the linear speedup observation in (19) as the $nT$ factor dominates the error of $||\nabla F(\mathbf{x})||^2$. Note that the solution stationarity with $n=40$ is slightly higher as expected from (37) that some higher order error may still remain in the bound of $||\nabla F(\mathbf{x})||^2$.

---

### Official Review · Reviewer_H1MF · 2025-07-03

**Clarity:** 2
**Significance:** 2
**Originality:** 2
**Rating:** 3
**Confidence:** 5

**Summary:**

This paper studies the decentralized optimization problem over unreliable and time-varying networks. It introduces a new stochastic primal-dual framework (FSPDA) that handles communication randomness through a stochastic augmented Lagrangian. Two algorithms are proposed, FSPDA-SA algorithm that adapts gradient updates to network conditions and FSPDA-STORM algorithm that adds variance reduction for better efficiency. Theoretical results show convergence rates of $O(1/\sqrt{T})$ for FSPDA-SA algorithm and $O(1/T^{2/3})$ for FSPDA-STORM algorithm, and experiments demonstrate their practical effectiveness.

**Questions:**

1. Could the authors clarify the benefits of formulating an augmented Lagrangian in the context of decentralized optimization? What advantages does this offer over standard formulations?

2. Are there any unique techniques employed in the convergence analysis for time-varying networks, as compared to the analysis typically used for static network settings?

3. Could the authors provide a more detailed discussion on how sparsification impacts the overall convergence rate? The experimental results suggest that increased sparsification may lead to slower and less stable convergence.

**Ethical Concerns:**

["NO or VERY MINOR ethics concerns only"]

**Final Justification:**

I have read all discussions and still hold concerns on the novelty and technical contributions in the work (please see the discussion with other reviewers for more details) so decided not to raise the score.

**Limitations:**

Please see above.

**Quality:**

2

**Strengths And Weaknesses:**

Strengths:
The main value in this work lies in the observation that randomness in time varying topolgy can be incorporated in a stochastic augmented Lagrangian formulation, which is an extension based on [Chang et al., 2020].

Weaknesses:
1. The problem setting addressed in the paper is somewhat limited and less aligned with current research interests. The work could be strengthened by extending the analysis to directed or B-connected time-varying networks, which are more general and practically relevant. As it stands, focusing solely on stochastic optimization over undirected time-varying networks narrows its applicability.

2. The proposed algorithms, FSPDA-SA and FSPDA-STORM, appear to be straightforward extensions of existing decentralized stochastic methods, incorporating SA and STORM components. This may be perceived as an incremental or relatively minor modification.

3. The convergence rates derived in the paper align with what is typically expected under the standard assumptions used in the paper and do not represent a significant theoretical advancement.

Overall, the work lacks substantial novelty and does not demonstrate particularly strong contributions relative to the current state of the field.

---

> ### Author Rebuttal · Authors · 2025-07-30
>
> Thank you for the critical and constructive comments. We wish to emphasize that the FSPDA algorithms work for a general setting of practical interests, and is accompanied with a novel analysis as well as potential to extend further. Please find our responses below.
>
> > The problem setting addressed in the paper is somewhat limited and less aligned with current research interests. The work could be strengthened by extending the analysis to directed or B-connected time-varying networks, which are more general and practically relevant. As it stands, focusing solely on stochastic optimization over undirected time-varying networks narrows its applicability.
> > The proposed algorithms, FSPDA-SA and FSPDA-STORM, appear to be straightforward extensions of existing decentralized stochastic methods, incorporating SA and STORM components. This may be perceived as an incremental or relatively minor modification.
>
> Compared to the settings suggested by the reviewer, we believe that our problem setting on **random time varying (TV) graphs** is not less practical and possibly more relevant to today's challenges. First, to clarify, our setting (also studied in [Lobel & Ozdaglar, 2010]) do not require each of the TV graphs ${\cal G}(\xi_a^t), t \geq 0$ to be connected - in the extreme case agents only need to communicate on a 1-edge random graph at each iteration. The graphs are only required to be connected in expectation (Assumption 3.2), which is similar to the B-connected TV graphs setting suggested by the reviewer. On top of this, we allow for random sparsification for further efficient communication and demonstrates that FSPDA-SA/STORM can be implemented as asynchronous algorithms. To our knowledge, the latter extensions may not be straightforward in the B-connected TV graph settings. We note that algorithms of directed graph requires very different strategies (e.g., push-pull techniques) and it remains an open problem for the analysis on non-convex optimization with PD-based algorithm under TV graph.
>
> In terms of algorithms development, both FSPDA-SA/STORM can be understood as "plug-n-play" derivatives of SA for finding the saddle point(s) of (4), obtained from analyzing the **stochastic linear constrained** version of the distributed learning problem (1). Together with the stochastic augmented Lagrangian function of the latter, these are novel perspectives offered by our paper and they naturally lead to decentralized algorithms on random TV graphs with random sparsification. In fact, we believe that the FSPDA framework based on (4) can provide a general recipe for designing more powerful algorithms than FSPDA-SA/STORM. The analysis for these algorithms also prompted us to design new proof techniques.
>
> > The convergence rates derived in the paper align with what is typically expected under the standard assumptions used in the paper and do not represent a significant theoretical advancement.
>
> While we agree that the rates are as expected, as depicted in Table 1, we emphasize that our paper is the first one to achieve these rates under the settings of TV graphs without relying on the common bounded heterogeneity (BH) assumption, which has caused slow convergence in practice for certain prior works such as DSGD. Although it appears to be unsurprising, this is a crucial step in extending decentralized optimization towards practical scenarios.
>
> > Could the authors clarify the benefits of formulating an augmented Lagrangian in the context of decentralized optimization? What advantages does this offer over standard formulations?
>
> Our augmented Lagrangian formulation incorporates a novel stochastic equality constraint to enforce/promote consensus. As demonstrated in the discussions after (4), it enables one to derive algorithms that **naturally** supports communication on random TV graphs with sparsification. Another advantage is that the approach naturally gives rise (via stochastic gradient descent/ascent) to stable algorithms with strong convergence guarantees, e.g., without the bounded heterogeneity assumption, linear convergence under PL condition with deterministic gradient, etc.
>
> On the other hand, algorithms derived under the FSPDA framework can be seen as TV graph extensions of gradient tracking, EXTRA, etc., see Sec. 2.1 and [Chang et al., 2020]. Note it has been pointed out that the analysis of TV graph GT algorithm is not straight-forward as it involves the communication of two different variables and therefore the non-symmetric product of two mixing matrices [Koloskova et al., 2021]. This brings greater flexibility and stronger guarantees than algorithms from standard approach such as the primal-only DSGD algorithms.
>
>
> > Are there any unique techniques employed in the convergence analysis for time-varying networks, as compared to the analysis typically used for static network settings?
>
> There are several unique techniques deployed in our analysis due to the non-triviality in handling TV graphs. In particular, instead of the natural way (as in [Hong et al., 2017]) of designing a Lyapunov function based on the expected Lagrangian, we view the convergence of FSPDA(-SA/STORM) as that of stochastic approximation scheme on a Lyapunov function that is dependent on the global objective $F(x)$ and the system error quantities such as consensus error and dual gradient tracking error, see Section 3.1. As shown in Appendix C, we control/introduce the error terms in a step-by-step fashion. The application of Assumption 3.4 is particularly unique in the proof of Lemma C.2. This assumption treats random graphs as stochastic estimation to the graph Laplacian operator $\mathbf{A}^\top \mathbf{A} \mathbf{x}$ and enables us to relate the random graph error to a quantity proportional to the consensus error $|| \mathbf{x} ||_{\mathbf{K}}^2$.
>
> > Could the authors provide a more detailed discussion on how sparsification impacts the overall convergence rate? The experimental results suggest that increased sparsification may lead to slower and less stable convergence.
>
> The reviewer is correct that sparsification will lead to slower convergence and less stable convergence. This is an inevitable tradeoff. Our analysis, however, can shed light on what is the right degree of sparsification to be applied. To set this up, we examine the quantities that are dependent on the degree of sparsification. They are $\rho_{\min}$ in (13) and $\sigma_A$ in (15). In particular, when the degree of sparsification increases which reduces the magnitude ${\bf R}$, it is easy to see that $\rho_{\min}$ decreases and $\sigma_A^2$ increases.
>
> We focus on FSPDA-SA for illustration purpose. From Theorem 3.5/Theorem C.5 and the discussions that follow, we see that the effects of $\rho_{\min}, \sigma_A$ vanish as $T \to \infty$. In other words, the level of sparsification only affect the performance in the *transient* stages (when $T$ is not large). At a closer examination, in (50), we see that the transient terms depending on $\rho_{\min}, \sigma_A$ get multiplied by $L^2$. From here, we conclude that when $L$ is large, i.e., when the problem is less smooth (e.g., it involves training an NN with large no of layers), choosing a high sparsification level can be detrimental. We will include a discussion on the above matters in the final version.
>
> We look forward to discussing with the reviewer if any further clarifications are needed during the discussion period.

---

> > ### Comment · Reviewer_H1MF · 2025-08-01
> >
> > I appreciate the authors’ efforts in addressing the concerns. Please see the follow-up comments below:
> >
> > 1.	Regarding the connectivity assumption,
> > it is typically hard for distributed optimization algorithm to derive the average consensus in the analysis, especially for time-varying graphs. However, from Assumption 2 it seems that the derivation of average consensus will be trivial and this could diminish the analytical challenge that typically arises in decentralized optimization over time-varying graphs.
> >
> > 2.	“On top of this, we allow for random sparsification for further efficient communication” -
> > I appreciate the authors' discussion on the incorporation of random sparsification for communication efficiency. I would like to point out that similar ideas have been explored in [*Communication-efficient variance-reduced decentralized stochastic optimization over time-varying directed graphs." IEEE Transactions on Automatic Control 67.12 (2021)*], which also addresses communication reduction via sparsification in a time-varying and directed setting. It would strengthen the contribution of this paper if the authors could elaborate on what new theoretical insights or practical behaviors emerge in their current analysis that go beyond existing results.
> >
> > 3.	Regarding the scope of Time-Varying Graph settings,
> > The current work focuses on undirected time-varying graphs. While this is a meaningful setting, I encourage the authors to consider that recent work has tackled the more general and challenging setting of time-varying directed networks. These settings are often closer to real-world decentralized systems (please see papers listed below). Given the maturity of the literature in that area, the exclusion of directed topologies may limit the broader applicability and novelty of the current work. Including a discussion on this point would help clarify the scope and relevance of the contributions.
> >
> > a).	*"Accelerated distributed stochastic non-convex optimization over time-varying directed networks." IEEE Transactions on Automatic Control (2024)*.
> >
> > b).	*"Accelerated $ AB $/Push–Pull Methods for Distributed Optimization Over Time-Varying Directed Networks." IEEE Transactions on Control of Network Systems 11.3 (2023)*.
> >
> > c).	*"AB/Push-Pull method for distributed optimization in time-varying directed networks." Optimization Methods and Software (2023)*.
> >
> > d).	*"Decentralized optimization over time-varying directed graphs with row and column-stochastic matrices." IEEE Transactions on Automatic Control 65.11 (2020)*.
> >
> > 4.	“ we emphasize that our paper is the first one to achieve these rates under the settings of TV graphs without relying on the common bounded heterogeneity (BH) assumption, which has caused slow convergence in practice for certain prior works such as DSGD.” -
> > Please note that the paper listed above (*Accelerated distributed stochastic non-convex optimization over time-varying directed networks.*) has established same convergence rate for non-convex problems over time-varying directed networks. I would encourage the authors to more carefully position their claims relative to these recent developments, potentially clarifying what specific aspects of their analysis (e.g., in algorithm design, proof techniques, or assumptions) distinguish their results from prior work.

---

### Note · Authors · 2025-08-15

Dear Area Chair and Reviewers,

We sincerely thank the AC and reviewers for the constructive discussions during the rebuttal period. We would like to take this final chance of communication to summarize a few points from our discussions.

- This paper proposed the FSPDA algorithms for decentralized stochastic optimization over **random (time-varying)** networks with **fast convergence** for possibly non-convex problems. Besides showing convergence under a relaxed set of assumptions: (a) free of bounded heterogeneity, (b) arbitrarily random TV graphs, FSPDA also supports for local updates, and admits improved iteration complexity (via the FSPDA-STORM variant), and thus achieving comparable performance to SOTA communication efficient algorithms that require stronger conditions for implementation/convergence. We believe that these technical contributions have been commended favorably by reviewers **RPmm** and **Rmsh**.
- During the disucssion period, we resolved many concerns of reviewer **TNhj** on the feasibility of FSPDA's stepsize conditions. Particularly, our stepsizes are feasible as most practical graphs are sparse with a finite number of edges and similar requirements can be found in the literature.
- Since we did not manage to engage reviewer **H1MF** for further discussion after the initial exchange, it remains uncertain if the concern of the reviewer on the comparison with prior works on time varying directed graphs has been resolved. In our response, we pointed out that the existing literature suggested by the reviewer either requires stronger assumptions on **the type of TV graphs**, or **strongly convex problem**. Both restrictions are not present for FSPDA. Nevertheless, we agree that there remains an exciting open problem to extend the design principle of FSPDA to TV directed graphs.
- Reviewer **RPmm** is concerned with the performance of our algorithm on large graphs. We will include a set of experiments on larger networks to illustrate the effect of graph size on the practical performance of our algorithm.
- The revision will incorporate the reviewers' suggestions on clarity, literature survey, and additional experiments on large graphs, etc. Thank you for the valuable feedback!

We hope this summary wraps up our discussion and gives a clear picture for the AC and reviewers to make a decision on our submission.

Best regards,

Authors of Paper 26853.

---

### Decision · Program_Chairs · 2025-09-17

**Decision:**

Reject

**Comment:**

This work proposes an algorithm for decentralized training of neural networks in a time-varying setting. Unlike prior works such as Kovalev et al., which explicitly assume strong convexity, this paper relaxes the assumption to smooth functions. The contribution includes experimental results and convergence bounds, which are of interest, though in my opinion, their generality and relation to the state of the art are limited and not sufficiently contrasted.

The reviewers appreciated the effort to present a rigorous framework for the non convex loss time varying communication matrix setting. However, the formulation is very similar to Kovalev et al., as it also relies on a Lagrangian approach and saddle-point condition. Moreover, the lack of discussion of key related work, e.g., Boyd et al., Randomized Gossip Algorithms, is surprising and weakens the positioning of the contribution. Concerns were raised, in line with reviewer H1MF, regarding the novelty of the work. For a resubmission, addressing these missing references and clarifying the distinct contribution will be necessary, for instance by summarizing and contrasting the differences more precisely (as highlighted by the authors during the rebuttal).

During the rebuttal phase, three reviewers were convinced of the paper’s merits, notably its attempt to provide a framework for a specific problem. However, one reviewer maintained strong concerns about the incomplete literature review. I find this objection persuasive: while the problem is interesting, the related work discussion is insufficiently developed (it's solely done line 29-54 instead of a full section), which prevents acceptance at this stage, given the competitiveness of this specific area.

Thus, I recommend rejection given the substantial rewriting needed.